# PAIRWISE CONFIDENCE DIFFERENCE ON UNLABELED DATA IS SUFFICIENT FOR BINARY CLASSIFICATION

## ABSTRACT

Learning with confidence labels is an emerging weakly supervised learning paradigm, where training data are equipped with *confidence labels* instead of *exact labels*. Positive-confidence (Pconf) classification is a typical learning problem in this context, where we are given only positive data equipped with confidence. However, pointwise confidence may not be accessible in real-world scenarios. In this paper, we dive into a novel weakly supervised learning problem called confidence-difference (ConfDiff) classification. Instead of pointwise confidence, we are given only unlabeled data pairs equipped with *confidence difference* specifying *the difference in the probabilities of being positive*. An unbiased risk estimator is derived to tackle the problem, and we show that the estimation error bound achieves the optimal convergence rate. Extensive experiments on benchmark data sets validate the effectiveness of our proposed approaches in leveraging the supervision information of the confidence difference.

## 1 INTRODUCTION

Recent years have witnessed the prevalence of deep learning and its successful applications. However, the success is built on the basis of the collection of large amounts of data with unique and accurate labels. In many real-world scenarios, it is often difficult to satisfy such requirements. To circumvent the difficulty, various weakly supervised learning problems have been investigated accordingly, including but not limited to semi-supervised learning (Chapelle et al., 2006; Zhu & Goldberg, 2009; Li & Zhou, 2015; Berthelot et al., 2019), label-noise learning (Patrini et al., 2017; Han et al., 2018; Li et al., 2021; Wang et al., 2021; Wei et al., 2022), positive-unlabeled learning (du Plessis et al., 2014; Su et al., 2021; Yao et al., 2022), partial-label learning (Cour et al., 2011; Wang & Zhang, 2020; Wen et al., 2021; Wang et al., 2022; Wu et al., 2022), unlabeled-unlabeled learning (Lu et al., 2019; 2020) and similarity-based classification (Bao et al., 2018; Cao et al., 2021b; Bao et al., 2022).

Learning with confidence labels (Ishida et al., 2018; Cao et al., 2021a;b) is another weakly supervised learning paradigm, where we are given training examples with *confidence labels* instead of *exact labels*. Positive-confidence (Pconf) classification (Ishida et al., 2018) is a problem setting within this scope, which is aimed at learning a binary classifier from only positive data equipped with confidence (the probability of being positive) without negative data. Pconf classification can alleviate the difficulty when negative data cannot be acquired due to privacy or security issues during the data annotation process. The need to learn from such inexact supervision widely exists in real-world scenarios, such as purchase prediction (Ishida et al., 2018), user preservation prediction (Ishida et al., 2018), drivers' drowsiness prediction (Shinoda et al., 2020), etc.

However, the process of collecting large amounts of training examples with pointwise confidence might be actually demanding under many circumstances, since it is tough to describe the probability of being positive for each training example exactly (Shinoda et al., 2020). Feng et al. (2021) showed that *learning from pairwise comparisons* could serve as an alternative strategy given limited pointwise labeling information. Inspired by it, we investigate a more practical problem setting in this paper, where we are given only *unlabeled data pairs with confidence difference* indicating the difference in the probabilities of being positive. Compared with pointwise confidence, confidence difference can be collected more easily in many real-world scenarios. Take click-through rate prediction in recommender systems (Zhang et al., 2019) for example. The combinations of users and

their favorite/disliked items can be regarded as positive/negative data. When collecting training data, it is not easy to distinguish between positive and negative data. Furthermore, the positive confidence of training data may be difficult to be determined due to the extremely sparse and class-imbalance problems (Yao et al., 2021). However, it is much easier to obtain the difference in the preference between a pair of candidate items for a given user. Take the disease risk estimation problem for another example. The goal is to predict the risk of having some disease given a person's attributes. When asking doctors to annotate the probabilities of having the disease for people, it is not easy to determine the exact values of the probabilities. Furthermore, the probability values given by different doctors may be different due to personally subjective assumptions and will deviate from the ground-truth values. However, it is much easier and less biased to estimate the relative difference in the probabilities of having the disease between two people.

Our contributions are summarized as follows:

- We investigate confidence-difference (ConfDiff) classification, a novel and practical weakly supervised learning problem, which can be solved via *empirical risk minimization* by constructing an *unbiased risk estimator*. The proposed approach can be equipped with any model, loss function, and optimizer flexibly.
- The estimation error bound is derived, showing that the proposed approach achieves the optimal parametric convergence rate. The robustness is further demonstrated by probing into the influence of an inaccurate class prior probability and noisy confidence difference.
- To mitigate overfitting issues, a risk correction approach (Lu et al., 2020) with consistency guarantee is further introduced. Extensive experimental results on benchmark data sets validate the effectiveness of the proposed approaches.

**Related works.** Learning with pairwise comparisons has been investigated pervasively in the community (Burges et al., 2005; Cao et al., 2007; Jamieson & Nowak, 2011; Park et al., 2015; Kane et al., 2017; Xu et al., 2017; Shah et al., 2019), with applications in information retrieval (Liu, 2011), computer vision (Fu et al., 2015), regression (Xu et al., 2019; 2020), crowdsourcing (Chen et al., 2013; Zeng & Shen, 2022), graph learning (He et al., 2022), etc. It is noteworthy that there exist distinct differences between our work and previous works on learning with pairwise comparisons. Previous works have mainly tried to learn a ranking function which can rank candidate examples according to the relevance or preference. In this paper, we try to learn a *pointwise binary classifier* by conducting empirical risk minimization under the binary classification setting.

**Relationship to Pcomp classification.** Feng et al. (2021) elaborated that a binary classifier could be learned from pairwise comparisons, which was termed as Pcomp classification. There are distinct differences between our work and Pcomp classification. First, Pcomp classification is not capable of leveraging the fine-grained confidence difference, which can be incidentally obtained when collecting pairwise comparison data. We will experimentally elucidate the benefit of exploiting the confidence difference in the later section. Second, the assumptions of the data generation process are different. Pcomp classification assumes that the unlabeled data pair is *ordered*, where the first instance is more likely to be positive than the other. In ConfDiff classification, the instances of the unlabeled data pair are *independent*, which can be easier to collect.

## 2 PRELIMINARIES

In this section, we introduce the notations used in this paper and discuss the background of binary classification, Pconf classification and Pcomp classification. Then, we elucidate the data generation process of confidence-difference classification.

### 2.1 BINARY CLASSIFICATION

For binary classification, let $\mathcal{X} = \mathbb{R}^d$ denote the $d$-dimensional feature space and $\mathcal{Y} = \{+1, -1\}$ denote the label space. Let $p(\boldsymbol{x}, y)$ denote the unknown joint probability distribution over random variables $(\boldsymbol{x}, y) \in \mathcal{X} \times \mathcal{Y}$. The task of binary classification is to learn a binary classifier $g : \mathcal{X} \to \mathbb{R}$ which minimizes the following classification risk:

$$R(g) = \mathbb{E}_{p(\boldsymbol{x},y)}[\ell(g(\boldsymbol{x}), y)], \qquad (1)$$

where $\ell(\cdot, \cdot)$ is a non-negative binary-class loss function, such as the 0-1 loss and logistic loss. Let $\pi_+ = p(y = +1)$ and $\pi_- = p(y = -1)$ denote the class prior probabilities for the positive and negative classes respectively. Furthermore, let $p_+(\boldsymbol{x}) = p(\boldsymbol{x}|y = +1)$ and $p_-(\boldsymbol{x}) = p(\boldsymbol{x}|y = -1)$ denote the class-conditional probability densities of positive and negative data respectively. Then the classification risk in Eq. (1) can be equivalently expressed as

$$R(g) = \pi_+ \mathbb{E}_{p_+(\boldsymbol{x})}[\ell(g(\boldsymbol{x}), +1)] + \pi_- \mathbb{E}_{p_-(\boldsymbol{x})}[\ell(g(\boldsymbol{x}), -1)]. \tag{2}$$

## 2.2 Positive-Confidence (Pconf) classification

In many real-world applications, it may be difficult to collect negative data. Pconf classification (Ishida et al., 2018) is aimed at inducing a binary classifier from only positive data. The additional requirement is that the confidence of being positive should be accessible to the learning algorithm. Given only positive data equipped with confidence $\{(\boldsymbol{x}_i, r_i)\}_{i=1}^n$, Ishida et al. (2018) provided an unbiased risk estimator to conduct empirical risk minimization:

$$\widehat{R}_{\mathrm{Pconf}}(g) = \frac{\pi_+}{n} \sum_{i=1}^n (\ell(g(\boldsymbol{x}_i), +1) + \frac{1 - r_i}{r_i} \ell(g(\boldsymbol{x}_i), -1)), \tag{3}$$

where $r_i = p(y_i = +1|\boldsymbol{x}_i)$ is the positive confidence associated with $\boldsymbol{x}_i$. However, pointwise positive confidence may not be easy to obtain in real-world scenarios (Shinoda et al., 2020).

## 2.3 Pairwise-Comparison (PComp) classification

Pcomp classification is a weakly supervised binary classification problem (Feng et al., 2021). In Pcomp classification, we are given pairs of unlabeled data where we know which one is more likely to be positive than the other. It is assumed that Pcomp data are sampled from labeled data pairs whose labels belong to $\{(+1, -1), (+1, +1), (-1, -1)\}$. Based on this assumption, the probability density of Pcomp data $(\boldsymbol{x}, \boldsymbol{x}')$ is given as

$$\widetilde{p}(\boldsymbol{x}, \boldsymbol{x}') = \frac{q(\boldsymbol{x}, \boldsymbol{x}')}{\pi_+^2 + \pi_-^2 + \pi_+ \pi_-}, \tag{4}$$

where $q(\boldsymbol{x}, \boldsymbol{x}') = \pi_+^2 p_+(\boldsymbol{x}) p_+(\boldsymbol{x}') + \pi_-^2 p_-(\boldsymbol{x}) p_-(\boldsymbol{x}') + \pi_+ \pi_- p_+(\boldsymbol{x}) p_-(\boldsymbol{x}')$. Then, an unbiased risk estimator for Pcomp classification is derived as follows:

$$\widehat{R}_{\mathrm{Pcomp}}(g) = \frac{1}{n} \sum_{i=1}^n (\ell(g(\boldsymbol{x}_i), +1) + \ell(g(\boldsymbol{x}'_i), -1) - \pi_+ \ell(g(\boldsymbol{x}_i), -1) - \pi_- \ell(g(\boldsymbol{x}'_i), +1)). \tag{5}$$

In real-world applications, we may not only know one example is more likely to be positive than the other, but also know how much *the difference of confidence* is. Next, a novel weakly supervised learning setting named ConfDiff classification is introduced.

## 2.4 Confidence-Difference (ConfDiff) classification

In this subsection, the formal definition of confidence difference is given firstly. Then, we elaborate the data generation process of ConfDiff data.

**Definition 1** (Confidence Difference). *The confidence difference $c(\boldsymbol{x}, \boldsymbol{x}')$ between the unlabeled data pair $(\boldsymbol{x}, \boldsymbol{x}')$ is defined as*

$$c(\boldsymbol{x}, \boldsymbol{x}') = p(y' = 1|\boldsymbol{x}') - p(y = 1|\boldsymbol{x}). \tag{6}$$

As shown in the definition above, the confidence difference denotes the difference in the class posterior probabilities between the unlabeled data pair, which can measure how confident the pairwise comparison is. In ConfDiff classification, we are only given $n$ unlabeled data pairs with confidence difference $\mathcal{D} = \{((\boldsymbol{x}_i, \boldsymbol{x}'_i), c_i)\}_{i=1}^n$. Here, $c_i = c(\boldsymbol{x}_i, \boldsymbol{x}'_i)$ is the confidence difference for the unlabeled data pair $(\boldsymbol{x}_i, \boldsymbol{x}'_i)$. Furthermore, the unlabeled data pair $(\boldsymbol{x}_i, \boldsymbol{x}'_i)$ is assumed to be drawn from a probability density $p(\boldsymbol{x}, \boldsymbol{x}') = p(\boldsymbol{x})p(\boldsymbol{x}')$. This indicates that $\boldsymbol{x}_i$ and $\boldsymbol{x}'_i$ are two i.i.d. instances sampled from $p(\boldsymbol{x})$. It is worth noting that the confidence difference $c_i$ will be positive if the second instance $\boldsymbol{x}'_i$ has a higher probability to be positive than the first instance $\boldsymbol{x}_i$, and will be negative otherwise. During the data collection process, the labeler can first sample two unlabeled data from the marginal distribution $p(\boldsymbol{x})$, then provide the confidence difference for them. This data generation assumption makes the unlabeled data pairs easier to be collected.

# 3 THE PROPOSED APPROACH

In this section, an unbiased risk estimator is presented for ConfDiff classification. Then, we give an estimation error bound to show the convergence property. Besides, we show the influence of an inaccurate class prior probability and noisy confidence difference on the risk estimator. Furthermore, a risk correction approach (Lu et al., 2020) is elaborated to improve the generalization performance of our proposed approach.

## 3.1 UNBIASED RISK ESTIMATOR

In this subsection, we show that the classification risk in Eq. (1) can be expressed with ConfDiff data in the equivalent way.

**Theorem 1.** *The classification risk $R(g)$ in Eq. (1) can be equivalently expressed as*

$$R_{\mathrm{CD}}(g) = \mathbb{E}_{p(\boldsymbol{x}, \boldsymbol{x}')}[\frac{1}{2}(\mathcal{L}(\boldsymbol{x}, \boldsymbol{x}') + \mathcal{L}(\boldsymbol{x}', \boldsymbol{x}))], \tag{7}$$

*where*

$$\mathcal{L}(\boldsymbol{x}, \boldsymbol{x}') = (\pi_+ - c(\boldsymbol{x}, \boldsymbol{x}'))\ell(g(\boldsymbol{x}), +1) + (\pi_- - c(\boldsymbol{x}, \boldsymbol{x}'))\ell(g(\boldsymbol{x}'), -1).$$

Accordingly, we can derive an unbiased risk estimator for ConfDiff classification:

$$\widehat{R}_{\mathrm{CD}}(g) = \frac{1}{2n} \sum_{i=1}^{n}((\pi_+ - c_i)\ell(g(\boldsymbol{x}_i), +1) + (\pi_- - c_i)\ell(g(\boldsymbol{x}'_i), -1)$$
$$+ (\pi_+ + c_i)\ell(g(\boldsymbol{x}'_i), +1) + (\pi_- + c_i)\ell(g(\boldsymbol{x}_i), -1)). \tag{8}$$

To estimate the class prior probability $\pi_+$, we can transform ConfDiff data into Pcomp data by ranking the two instances in the unlabeled data pair according to the confidence difference. Then, we can adopt the approach proposed in Feng et al. (2021) to estimate $\pi_+$. It is worth noting that the risk estimator in Eq. (3) for Pconf classification is very sensitive to small confidence values, while our risk estimator will not be influenced by them.

**Minimum-variance risk estimator.** Actually, Eq. (8) is one of the candidates of the unbiased risk estimator. We introduce the following lemma:

**Lemma 1.** *The following expression is also an unbiased risk estimator:*

$$\frac{1}{n} \sum_{i=1}^{n}(\alpha\mathcal{L}(\boldsymbol{x}_i, \boldsymbol{x}'_i) + (1 - \alpha)\mathcal{L}(\boldsymbol{x}'_i, \boldsymbol{x}_i)), \tag{9}$$

*where $\alpha \in [0, 1]$ is an arbitrary weight.*

Then, we introduce the following theorem:

**Theorem 2.** *The unbiased risk estimator in Eq. (8) has the minimum variance among all the candidate unbiased risk estimators in the form of Eq. (9) w.r.t. $\alpha \in [0, 1]$.*

Theorem 2 indicates the variance minimality of the proposed unbiased risk estimator in Eq. (8), and we adopt this risk estimator in the following sections.

## 3.2 ESTIMATION ERROR BOUND

In this subsection, we elaborate the convergence property of the proposed risk estimator $\widehat{R}_{\mathrm{CD}}(g)$ by giving an estimation error bound. Let $\mathcal{G} = \{g : \mathcal{X} \mapsto \mathbb{R}\}$ denote the model class. It is assumed that there exists some constant $C_g$ such that $\sup_{g \in \mathcal{G}} \|g\|_\infty \leq C_g$ and some constant $C_\ell$ such that $\sup_{|z| \leq C_g} \ell(z, y) \leq C_\ell$. We also assume that the binary loss function $\ell(z, y)$ is Lipschitz continuous for $z$ and $y$ with a Lipschitz constant $L_\ell$. [1] Let $g^* = \arg\min_{g \in \mathcal{G}} R(g)$ denote the minimizer of the classification risk in Eq. (1) and $\widehat{g}_{\mathrm{CD}} = \arg\min_{g \in \mathcal{G}} \widehat{R}_{\mathrm{CD}}(g)$ denote the minimizer of the unbiased risk estimator in Eq. (8). The following theorem can be derived:

---

[1]The theoretical analysis in the next subsections is also based on these assumptions. For simplicity, we do not restate them in the next subsections.

**Theorem 3.** *For any $\delta > 0$, the following inequality holds with probability at least $1 - \delta$:*

$$R(\widehat{g}_{\mathrm{CD}}) - R(g^*) \leq 8L_\ell \mathfrak{R}_n(\mathcal{G}) + 4C_\ell \sqrt{\frac{\ln 2/\delta}{2n}}, \tag{10}$$

*where $\mathfrak{R}_n(\mathcal{G})$ denotes the Rademacher complexity of $\mathcal{G}$ for unlabeled data with size $n$.*

From Theorem 3, we can observe that as $n \to \infty$, $R(\widehat{g}_{\mathrm{CD}}) \to R(g^*)$ because $\mathcal{R}_n(\mathcal{G}) \to 0$ for all parametric models with a bounded norm, such as deep neural networks trained with weight decay (Golowich et al., 2018). Furthermore, the estimation error bound converges in $\mathcal{O}_p(1/\sqrt{n})$, where $\mathcal{O}_p$ denotes the order in probability, which is the optimal parametric rate for empirical risk minimization without making additional assumptions (Mendelson, 2008).

### 3.3 ROBUSTNESS OF RISK ESTIMATOR

In the previous subsections, it was assumed that the class prior probability is known in advance or estimated accurately. In addition, it was assumed that the ground-truth confidence difference of each unlabeled data pair is accessible. However, these assumptions can rarely be satisfied in real-world scenarios, since the collection of confidence difference is inevitably injected with noise. In this subsection, we theoretically analyze the influence of an inaccurate class prior probability and noisy confidence difference on the learning procedure. Later in subsection 4.4, we will experimentally verify our theoretical findings.

Let $\bar{\mathcal{D}} = \{((\boldsymbol{x}_i, \boldsymbol{x}_i'), \bar{c}_i)\}_{i=1}^n$ denote $n$ unlabeled data pairs with noisy confidence difference, where $\bar{c}_i$ is generated by corrupting the ground-truth confidence difference $c_i$ with noise. Besides, let $\bar{\pi}_+$ denote the inaccurate class prior probability accessible to the learning algorithm. Furthermore, let $\bar{R}_{\mathrm{CD}}(g)$ denote the empirical risk calculated based on the inaccurate class prior probability and noisy confidence difference. Let $\bar{g}_{\mathrm{CD}} = \arg\min_{g \in \mathcal{G}} \bar{R}_{\mathrm{CD}}(g)$ denote the minimizer of $\bar{R}_{\mathrm{CD}}(g)$. Then, the theorem demonstrating an estimation error bound is given as follows:

**Theorem 4.** *Based on the assumptions above, for any $\delta > 0$, the following inequality holds with probability at least $1 - \delta$:*

$$R(\bar{g}_{\mathrm{CD}}) - R(g^*) \leq 16L_\ell \mathfrak{R}_n(\mathcal{G}) + 8C_\ell \sqrt{\frac{\ln 2/\delta}{2n}} + \frac{4C_\ell \sum_{i=1}^n |\bar{c}_i - c_i|}{n} + 4C_\ell |\bar{\pi}_+ - \pi_+|. \tag{11}$$

Theorem 4 indicates that the estimation error is bounded by twice the original bound in Theorem 3 with the mean absolute error of the noisy confidence difference and the inaccurate class prior probability. Furthermore, if $\sum_{i=1}^n |\bar{c}_i - c_i|$ has a sublinear growth rate with high probability and the class prior probability is estimated consistently, the risk estimator can be even consistent. It elaborates the robustness of the proposed approach.

### 3.4 RISK CORRECTION APPROACH

It is worth noting that the empirical risk in Eq. (8) may be negative due to negative terms, which is unreasonable because of the non-negative property of loss functions. This phenomenon will result in severe overfitting problems when complex models are adopted (Lu et al., 2020; Cao et al., 2021b; Feng et al., 2021). To circumvent this difficulty, we wrap the individual loss terms in Eq. (8) with *risk correction functions* proposed in Lu et al. (2020), such as the rectified linear unit (ReLU) function $f(z) = \max(0, z)$ and the absolute value function $f(z) = |z|$. In this way, the corrected risk estimator for ConfDiff classification can be expressed as follows:

$$\widetilde{R}_{\mathrm{CD}}(g) = \frac{1}{2n}(f(\sum_{i=1}^n (\pi_+ - c_i)\ell(g(\boldsymbol{x}_i), +1)) + f(\sum_{i=1}^n (\pi_- - c_i)\ell(g(\boldsymbol{x}_i'), -1))$$

$$+ f(\sum_{i=1}^n (\pi_+ + c_i)\ell(g(\boldsymbol{x}_i'), +1)) + f(\sum_{i=1}^n (\pi_- + c_i)\ell(g(\boldsymbol{x}_i), -1))). \tag{12}$$

**Theoretical analysis.** We assume that the risk correction function $f(z)$ is Lipschitz continuous with Lipschitz constant $L_f$. For ease of notation, let $\widehat{A}_g = \sum_{i=1}^n (\pi_+ - c_i)\ell(g(\boldsymbol{x}_i), +1)/2n, \widehat{B}_g =$

$\sum_{i=1}^n (\pi_- - c_i)\ell(g(\boldsymbol{x}_i'), -1)/2n, \widehat{C}_g = \sum_{i=1}^n (\pi_+ + c_i)\ell(g(\boldsymbol{x}_i'), +1)/2n, \widehat{D}_g = \sum_{i=1}^n (\pi_- + c_i)\ell(g(\boldsymbol{x}_i), -1)/2n$. From Lemma 3 in Appendix A, the values of $\mathbb{E}[\widehat{A}_g], \mathbb{E}[\widehat{B}_g], \mathbb{E}[\widehat{C}_g]$, and $\mathbb{E}[\widehat{D}_g]$ are non-negative. Therefore, we assume that there exist non-negative constants $a, b, c, d$ such that $\mathbb{E}[\widehat{A}_g] \geq a, \mathbb{E}[\widehat{B}_g] \geq b, \mathbb{E}[\widehat{C}_g] \geq c$, and $\mathbb{E}[\widehat{D}_g] \geq d$. Besides, let $\widetilde{g}_{\mathrm{CD}} = \arg\min_{g \in \mathcal{G}} \widetilde{R}_{\mathrm{CD}}(g)$ denote the minimizer of $\widetilde{R}_{\mathrm{CD}}(g)$. Then, Theorem 5 is provided to elaborate the bias and consistency of $\widetilde{R}_{\mathrm{CD}}(g)$.

**Theorem 5.** *Based on the assumptions above, the bias of the risk estimator $\widetilde{R}_{\mathrm{CD}}(g)$ decays exponentially as $n \to \infty$:*

$$0 \leq \mathbb{E}[\widetilde{R}_{\mathrm{CD}}(g)] - R(g) \leq 2(L_f + 1)C_\ell \Delta, \tag{13}$$

*where $\Delta = \exp\left(-2a^2 n/C_\ell^2\right) + \exp\left(-2b^2 n/C_\ell^2\right) + \exp\left(-2c^2 n/C_\ell^2\right) + \exp\left(-2d^2 n/C_\ell^2\right)$. Furthermore, with probability at least $1 - \delta$, we have*

$$|\widetilde{R}_{\mathrm{CD}}(g) - R(g)| \leq 2C_\ell L_f \sqrt{\frac{\ln 2/\delta}{2n}} + 2(L_f + 1)C_\ell \Delta. \tag{14}$$

Theorem 5 demonstrates that $\widetilde{R}_{\mathrm{CD}}(g) \to R(g)$ in $\mathcal{O}_p(1/\sqrt{n})$, which means $\widetilde{R}_{\mathrm{CD}}(g)$ is biased yet consistent. The estimation error bound of $\widetilde{g}_{\mathrm{CD}}$ is analyzed in Theorem 6.

**Theorem 6.** *Based on the assumptions above, for any $\delta > 0$, the following inequality holds with probability at least $1 - \delta$:*

$$R(\widetilde{g}_{\mathrm{CD}}) - R(g^*) \leq 8L_\ell \mathfrak{R}_n(\mathcal{G}) + 4C_\ell(L_f + 1)\sqrt{\frac{\ln 2/\delta}{2n}} + 4(L_f + 1)C_\ell \Delta. \tag{15}$$

Theorem 6 elucidates that as $n \to \infty$, $R(\widetilde{g}_{\mathrm{CD}}) \to R(g^*)$, since $\mathcal{R}_n(\mathcal{G}) \to 0$ for all parametric models with a bounded norm (Mohri et al., 2012) and $\Delta \to 0$. Furthermore, the estimation error bound converges in $\mathcal{O}_p(1/\sqrt{n})$, which is the optimal parametric rate for empirical risk minimization without additional assumptions (Mendelson, 2008).

## 4 EXPERIMENTS

In this section, we verify the effectiveness of our proposed approaches experimentally.

### 4.1 EXPERIMENTAL SETUP

We conducted experiments on benchmark data sets, including MNIST (LeCun et al., 1998), Kuzushiji-MNIST (Clanuwat et al., 2018), Fashion-MNIST (Xiao et al., 2017), and CIFAR-10 (Krizhevsky & Hinton, 2009). In addition, four UCI data sets (Dua & Graff, 2017) were used, including Optdigits, USPS, Pendigits, and Letter. Since the data sets were originally designed for multi-class classification, we manually partitioned them into binary classes. The detailed descriptions of data sets is illustrated in Appendix. For CIFAR-10, we used ResNet-34 (He et al., 2016) as the model architecture. For other data sets, we used a multilayer perceptron (MLP) with three hidden layers of width 300 equipped with the ReLU (Nair & Hinton, 2010) activation function and batch normalization (Ioffe & Szegedy, 2015). The logistic loss is utilized to instantiate the loss function $\ell(\cdot, \cdot)$. It is worth noting that confidence difference is given by labelers in real-world applications, while it was generated synthetically in this paper to facilitate comprehensive experimental analysis. We firstly trained a probabilistic classifier via logistic regression with ordinarily labeled data and the same neural network architecture. Then, we sampled unlabeled data in pairs at random, and generated the class posterior probabilities by inputting them into the probabilistic classifier. After that, we generated confidence difference for each pair of sampled data according to Definition 1.

In the experiments, we adopted the following variants of our proposed approaches: 1) ConfDiff-Unbiased, which denotes the method working by minimizing the unbiased risk estimator proposed in Eq. (8); 2) ConfDiff-ReLU, which denotes the method working by minimizing the corrected risk estimator proposed in Eq. (12) with the ReLU function as the risk correction function; 3) ConfDiff-ABS, which denotes the method working by minimizing the corrected risk estimator proposed in Eq. (12) with the absolute value function as the risk correction function. We compared our proposed

Table 1: Classification accuracy (mean±std) of each method on benchmark data sets with different class priors, where the best performance is shown in bold.

| Class Prior | Method | MNIST | Kuzushiji | Fashion | CIFAR-10 |
|---|---|---|---|---|---|
| | Pcomp-Unbiased | 0.761±0.017 | 0.637±0.052 | 0.737±0.050 | 0.776±0.023 |
| | Pcomp-ReLU | 0.800±0.000 | 0.800±0.000 | 0.800±0.000 | 0.800±0.000 |
| | Pcomp-ABS | 0.800±0.000 | 0.800±0.000 | 0.800±0.000 | 0.800±0.000 |
| $\pi_+ = 0.2$ | Pcomp-Teacher | 0.965±0.010 | 0.871±0.046 | 0.853±0.017 | 0.836±0.019 |
| | ConfDiff-Unbiased | 0.789±0.041 | 0.672±0.053 | 0.855±0.024 | 0.789±0.025 |
| | ConfDiff-ReLU | 0.968±0.003 | 0.860±0.017 | 0.964±0.004 | 0.844±0.020 |
| | ConfDiff-ABS | **0.975±0.003** | **0.898±0.003** | **0.965±0.002** | **0.862±0.015** |
| Class Prior | Method | MNIST | Kuzushiji | Fashion | CIFAR-10 |
| | Pcomp-Unbiased | 0.712±0.020 | 0.578±0.036 | 0.723±0.042 | 0.703±0.042 |
| | Pcomp-ReLU | 0.502±0.003 | 0.502±0.004 | 0.500±0.000 | 0.602±0.032 |
| | Pcomp-ABS | 0.842±0.012 | 0.727±0.006 | 0.851±0.012 | 0.583±0.018 |
| $\pi_+ = 0.5$ | Pcomp-Teacher | 0.893±0.014 | 0.782±0.046 | 0.903±0.016 | 0.779±0.016 |
| | ConfDiff-Unbiased | 0.911±0.046 | 0.712±0.046 | 0.896±0.036 | 0.720±0.024 |
| | ConfDiff-ReLU | 0.944±0.011 | 0.805±0.015 | 0.960±0.003 | 0.830±0.007 |
| | ConfDiff-ABS | **0.964±0.001** | **0.867±0.006** | **0.967±0.001** | **0.843±0.004** |
| Class Prior | Method | MNIST | Kuzushiji | Fashion | CIFAR-10 |
| | Pcomp-Unbiased | 0.799±0.005 | 0.671±0.029 | 0.813±0.029 | 0.737±0.022 |
| | Pcomp-ReLU | 0.910±0.031 | 0.775±0.022 | 0.897±0.023 | 0.851±0.010 |
| | Pcomp-ABS | 0.854±0.027 | 0.838±0.026 | 0.921±0.017 | 0.849±0.007 |
| $\pi_+ = 0.8$ | Pcomp-Teacher | 0.943±0.026 | 0.814±0.027 | 0.936±0.014 | 0.821±0.003 |
| | ConfDiff-Unbiased | 0.792±0.017 | 0.758±0.033 | 0.810±0.035 | 0.794±0.012 |
| | ConfDiff-ReLU | 0.970±0.004 | 0.886±0.009 | 0.970±0.002 | 0.851±0.012 |
| | ConfDiff-ABS | **0.983±0.002** | **0.915±0.001** | **0.975±0.002** | **0.874±0.011** |

approaches with the following approaches: 1) Pcomp-Unbiased, which denotes the method working by minimizing the unbiased risk estimator for Pcomp classification proposed in Feng et al. (2021); 2) Pcomp-ReLU, which denotes the risk correction approach for Pcomp classification with the ReLU function as the risk correction function; 3) Pcomp-ABS, which denotes the risk correction approach for Pcomp classification with the absolute value function as the risk correction function; 4) Pcomp-Teacher, which denotes the state-of-the-art approach improving the label-noise learning approach RankPruning (Northcutt et al., 2017) with consistency regularization.

The number of training epoches was set to 200 and we obtained the testing accuracy by averaging the results in the last 10 epoches. The detailed hyperparameters can be found in Appendix. To verify the effectiveness of our approaches under different class prior settings, we set $\pi_+ \in \{0.2, 0.5, 0.8\}$ for all the data sets. For ease of implementation, we assumed that the class prior $\pi_+$ was known for all the compared methods. We repeated the sampling-and-training procedure for five times, and the mean accuracy as well as the standard deviation were recorded.

## 4.2 EXPERIMENTAL RESULTS

**Benchmark data sets.** Table 1 reports detailed experimental results for all the compared methods on four benchmark data sets. Based on Table 1, we can draw the following conclusions: a) On all the cases of benchmark data sets, our proposed ConfDiff-ABS method achieves superior performance against all of the other compared approaches significantly, which validates the effectiveness of our approach in utilizing supervision information from confidence difference; b) Pcomp-Teacher achieves superior performance against all of the other Pcomp approaches by a large margin. The excellent performance benefits from the effectiveness of consistency regularization for weakly supervised learning problems (Berthelot et al., 2019; Li et al., 2020; Wu et al., 2022); c) The risk correction methods for ConfDiff classification, i.e. ConfDiff-ReLU and ConfDiff-ABS, achieve better performance against ConfDiff-Unbiased, which elaborates that the risk correction technique is advantageous; d) It is worth noting that the classification results of ConfDiff-ReLU and ConfDiff-ABS have smaller variances than ConfDiff-Unbiased. It demonstrates that the risk correction method can enhance the stability and robustness for ConfDiff classification.

Table 2: Classification accuracy (mean±std) of each method on UCI data sets with different class priors, where the best performance is shown in bold.

| Class Prior | Method | Optdigits | USPS | Pendigits | Letter |
|---|---|---|---|---|---|
| | Pcomp-Unbiased | 0.771±0.016 | 0.721±0.046 | 0.743±0.057 | 0.757±0.028 |
| | Pcomp-ReLU | 0.800±0.000 | 0.800±0.000 | 0.800±0.000 | 0.800±0.000 |
| | Pcomp-ABS | 0.800±0.001 | 0.800±0.000 | 0.800±0.000 | 0.800±0.000 |
| $\pi_+ = 0.2$ | Pcomp-Teacher | 0.901±0.023 | 0.894±0.023 | 0.928±0.019 | 0.883±0.006 |
| | ConfDiff-Unbiased | 0.831±0.078 | 0.840±0.078 | 0.865±0.079 | 0.732±0.053 |
| | ConfDiff-ReLU | 0.953±0.014 | 0.957±0.007 | 0.987±0.003 | 0.929±0.008 |
| | ConfDiff-ABS | **0.963±0.009** | **0.960±0.005** | **0.988±0.002** | **0.942±0.007** |
| Class Prior | Method | Optdigits | USPS | Pendigits | Letter |
| | Pcomp-Unbiased | 0.651±0.112 | 0.671±0.090 | 0.748±0.038 | 0.632±0.019 |
| | Pcomp-ReLU | 0.630±0.076 | 0.554±0.048 | 0.514±0.019 | 0.525±0.023 |
| | Pcomp-ABS | 0.787±0.031 | 0.814±0.018 | 0.793±0.017 | 0.748±0.031 |
| $\pi_+ = 0.5$ | Pcomp-Teacher | 0.890±0.009 | 0.860±0.012 | 0.883±0.018 | 0.864±0.024 |
| | ConfDiff-Unbiased | 0.917±0.006 | 0.936±0.010 | 0.945±0.052 | 0.755±0.041 |
| | ConfDiff-ReLU | 0.921±0.011 | 0.945±0.009 | 0.981±0.004 | 0.895±0.006 |
| | ConfDiff-ABS | **0.962±0.006** | **0.959±0.004** | **0.988±0.003** | **0.925±0.003** |
| Class Prior | Method | Optdigits | USPS | Pendigits | Letter |
| | Pcomp-Unbiased | 0.765±0.023 | 0.746±0.012 | 0.743±0.026 | 0.694±0.031 |
| | Pcomp-ReLU | 0.902±0.017 | 0.891±0.024 | 0.913±0.023 | 0.827±0.025 |
| | Pcomp-ABS | 0.894±0.019 | 0.879±0.009 | 0.911±0.009 | 0.870±0.006 |
| $\pi_+ = 0.8$ | Pcomp-Teacher | 0.918±0.007 | 0.933±0.023 | 0.903±0.008 | 0.872±0.011 |
| | ConfDiff-Unbiased | 0.886±0.037 | 0.803±0.042 | 0.892±0.096 | 0.748±0.015 |
| | ConfDiff-ReLU | 0.949±0.007 | 0.958±0.008 | 0.986±0.003 | 0.927±0.008 |
| | ConfDiff-ABS | **0.964±0.005** | **0.964±0.003** | **0.987±0.002** | **0.945±0.007** |

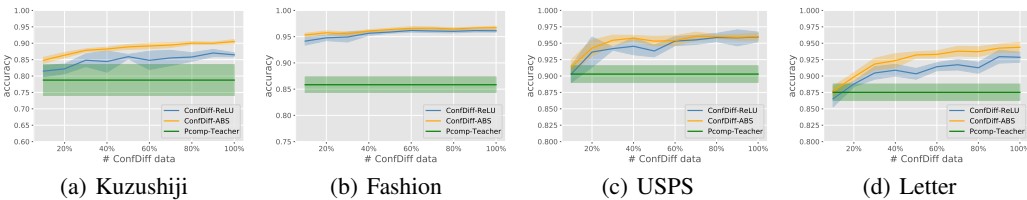

| (a) Kuzushiji | (b) Fashion | (c) USPS | (d) Letter |
|---|---|---|---|

Figure 1: Classification performance of ConfDiff-ReLU and ConfDiff-ABS given a fraction of training data as well as Pcomp-Teacher given 100% of training data ($\pi_+ = 0.2$).

**UCI data sets.** Table 2 reports detailed experimental results on four UCI data sets as well. From Table 2, we can observe that: a) On all the UCI data sets under different class prior probability settings, our proposed ConfDiff-ABS method achieves the best performance among all the compared approaches with significant superiority, which verifies the effectiveness of our proposed approaches again; b) The performance of our proposed approaches is more stable than the compared Pcomp approaches under different class prior probability settings, demonstrating the superiority of our methods in dealing with various kinds of data distributions; c) ConfDiff-Unbiased has comparable performance against its risk correction variants on some data sets while has inferior performance on some other data sets. This is mainly because some data sets have simpler patterns and are thus less affected by overfitting issues.

### 4.3 PERFORMANCE WITH FEWER TRAINING DATA

To validate the effectiveness of exploiting the confidence difference, we conducted experiments by changing the fraction of training data for ConfDiff-ReLU and ConfDiff-ABS (100% indicated that all the ConfDiff data were used for training). For comparison, we used 100% of training data for Pcomp-Teacher during the training process. Figure 1 shows the results on four data sets with $\pi_+ = 0.2$, and more experimental results can be found in Appendix. We can observe that the classification

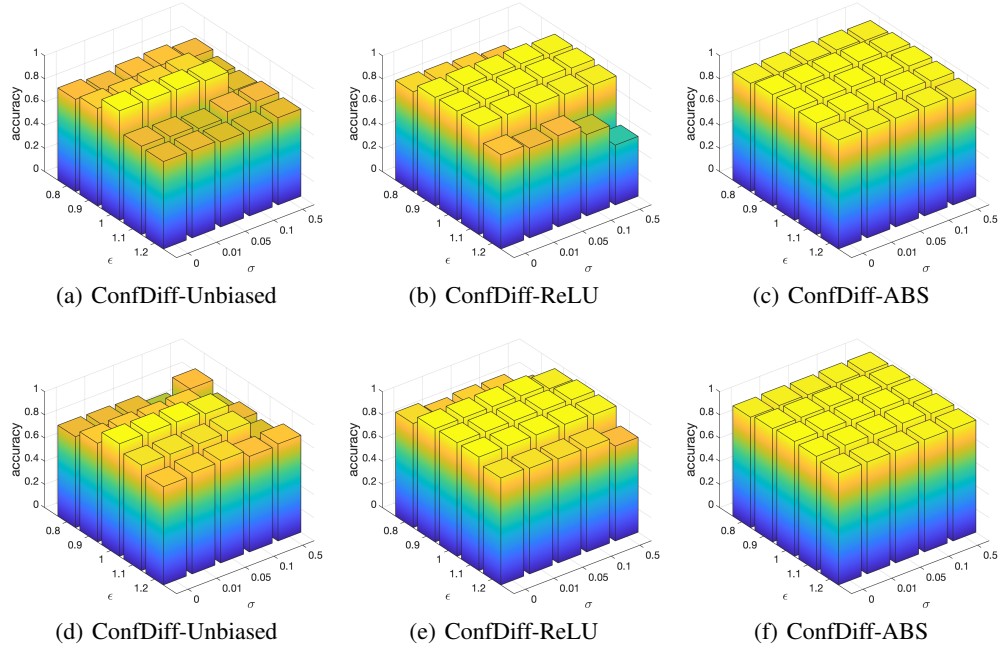

Figure 2: Classification accuracy on MNIST (the first row) and Pendigits (the second row) with $\pi_+ = 0.5$ given an inaccurate class prior probability and noisy confidence difference.

performance of our proposed approaches is still advantageous given a fraction of training data. Our approaches can achieve superior or comparable performance even when only 10% of training data are used. It validates the benefit and effectiveness of leveraging the supervision information of the confidence difference.

### 4.4 ANALYSIS ON ROBUSTNESS

In this subsection, we investigate the influence of an inaccurate class prior probability and noisy confidence difference on the generalization performance of the proposed approaches. Specifically, let $\bar{\pi}_+ = \epsilon \pi_+$ denote the corrupted class prior probability with $\epsilon$ being a real number around 1. Let $\bar{c}_i = \epsilon'_i c_i$ denote the noisy confidence difference where $\epsilon'_i$ is sampled from a normal distribution $\mathcal{N}(1, \sigma^2)$. Figure 2 shows the classification performance of our proposed approaches on MNIST and Pendigits ($\pi_+ = 0.5$) with different $\epsilon$ and $\sigma$. We can observe that ConfDiff-ABS is more robust against corruptions compared with ConfDiff-Unbiased and ConfDiff-ReLU. It is demonstrated that with $\bar{\pi}_+$ and $\bar{c}_i$ varying in a reasonable range, the performance is generally stable and even still superior against compared approaches. However, the performance degenerates with $\epsilon = 0.8$ or $\epsilon = 1.2$ on some data sets, which indicates that it is more important to obtain an accurate estimation of the class prior probability to facilitate model training.

## 5 CONCLUSION

In this paper, we dived into a novel weakly supervised learning setting where only unlabeled data pairs equipped with confidence difference were given. To solve the problem, an unbiased risk estimator was derived to perform empirical risk minimization. An estimation error bound was established to show that the optimal parametric convergence rate could be achieved. Furthermore, a risk correction approach was introduced to alleviate overfitting issues. Extensive experimental results validated the superiority of our proposed approaches. In future, it would be promising to apply our approaches in real-world scenarios.

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

# A    PROOF OF THEOREM 1

Before giving the proof of Theorem 1, we begin with the following lemmas:

**Lemma 2.** *The confidence difference $c(\boldsymbol{x}, \boldsymbol{x}')$ can be equivalently expressed as*

$$c(\boldsymbol{x}, \boldsymbol{x}') = \frac{\pi_+ p(\boldsymbol{x}) p_+(\boldsymbol{x}') - \pi_+ p_+(\boldsymbol{x}) p(\boldsymbol{x}')}{p(\boldsymbol{x}) p(\boldsymbol{x}')} \tag{16}$$

$$= \frac{\pi_- p_-(\boldsymbol{x}) p(\boldsymbol{x}') - \pi_- p(\boldsymbol{x}) p_-(\boldsymbol{x}')}{p(\boldsymbol{x}) p(\boldsymbol{x}')} \tag{17}$$

*Proof.* On one hand,

$$
\begin{aligned}
c(\boldsymbol{x}, \boldsymbol{x}') &= p(y' = 1|\boldsymbol{x}') - p(y = 1|\boldsymbol{x}) \\
&= \frac{p(\boldsymbol{x}', y' = 1)}{p(\boldsymbol{x}')} - \frac{p(\boldsymbol{x}, y = 1)}{p(\boldsymbol{x})} \\
&= \frac{\pi_+ p_+(\boldsymbol{x}')}{p(\boldsymbol{x}')} - \frac{\pi_+ p_+(\boldsymbol{x})}{p(\boldsymbol{x})} \\
&= \frac{\pi_+ p(\boldsymbol{x}) p_+(\boldsymbol{x}') - \pi_+ p_+(\boldsymbol{x}) p(\boldsymbol{x}')}{p(\boldsymbol{x}) p(\boldsymbol{x}')}.
\end{aligned}
$$

On the other hand,

$$
\begin{aligned}
c(\boldsymbol{x}, \boldsymbol{x}') &= p(y' = 1|\boldsymbol{x}') - p(y = 1|\boldsymbol{x}) \\
&= (1 - p(y' = 0|\boldsymbol{x}')) - (1 - p(y = 0|\boldsymbol{x})) \\
&= p(y = 0|\boldsymbol{x}) - p(y' = 0|\boldsymbol{x}') \\
&= \frac{p(\boldsymbol{x}, y = 0)}{p(\boldsymbol{x})} - \frac{p(\boldsymbol{x}', y = 0)}{p(\boldsymbol{x}')} \\
&= \frac{\pi_- p_-(\boldsymbol{x})}{p(\boldsymbol{x})} - \frac{\pi_- p_-(\boldsymbol{x}')}{p(\boldsymbol{x}')} \\
&= \frac{\pi_- p_-(\boldsymbol{x}) p(\boldsymbol{x}') - \pi_- p(\boldsymbol{x}) p_-(\boldsymbol{x}')}{p(\boldsymbol{x}) p(\boldsymbol{x}')},
\end{aligned}
$$

which concludes the proof.  $\square$

**Lemma 3.** *The following equations hold:*

$$\mathbb{E}_{p(\boldsymbol{x}, \boldsymbol{x}')}[(\pi_+ - c(\boldsymbol{x}, \boldsymbol{x}'))\ell(g(\boldsymbol{x}), +1)] = \pi_+ \mathbb{E}_{p_+(\boldsymbol{x})}[\ell(g(\boldsymbol{x}), +1)], \tag{18}$$

$$\mathbb{E}_{p(\boldsymbol{x}, \boldsymbol{x}')}[(\pi_- + c(\boldsymbol{x}, \boldsymbol{x}'))\ell(g(\boldsymbol{x}), -1)] = \pi_- \mathbb{E}_{p_-(\boldsymbol{x})}[\ell(g(\boldsymbol{x}), -1)], \tag{19}$$

$$\mathbb{E}_{p(\boldsymbol{x}, \boldsymbol{x}')}[(\pi_+ + c(\boldsymbol{x}, \boldsymbol{x}'))\ell(g(\boldsymbol{x}'), +1)] = \pi_+ \mathbb{E}_{p_+(\boldsymbol{x}')}[\ell(g(\boldsymbol{x}'), +1)], \tag{20}$$

$$\mathbb{E}_{p(\boldsymbol{x}, \boldsymbol{x}')}[(\pi_- - c(\boldsymbol{x}, \boldsymbol{x}'))\ell(g(\boldsymbol{x}'), -1)] = \pi_- \mathbb{E}_{p_-(\boldsymbol{x}')}[\ell(g(\boldsymbol{x}'), -1)]. \tag{21}$$

*Proof.* Firstly, the proof of Eq. (18) is given:

$$
\begin{aligned}
&\mathbb{E}_{p(\boldsymbol{x},\boldsymbol{x}')}[(\pi_+ - c(\boldsymbol{x},\boldsymbol{x}'))\ell(g(\boldsymbol{x}),+1)] \\
&= \int\int \frac{\pi_+ p(\boldsymbol{x})p(\boldsymbol{x}') - \pi_+ p(\boldsymbol{x})p_+(\boldsymbol{x}') + \pi_+ p_+(\boldsymbol{x})p(\boldsymbol{x}')}{p(\boldsymbol{x})p(\boldsymbol{x}')}\ell(g(\boldsymbol{x}),+1)p(\boldsymbol{x},\boldsymbol{x}')\,\mathrm{d}\boldsymbol{x}\,\mathrm{d}\boldsymbol{x}' \\
&= \int\int (\pi_+ p(\boldsymbol{x})p(\boldsymbol{x}') - \pi_+ p(\boldsymbol{x})p_+(\boldsymbol{x}') + \pi_+ p_+(\boldsymbol{x})p(\boldsymbol{x}'))\ell(g(\boldsymbol{x}),+1)\,\mathrm{d}\boldsymbol{x}\,\mathrm{d}\boldsymbol{x}' \\
&= \int \pi_+ p(\boldsymbol{x})\ell(g(\boldsymbol{x}),+1)\,\mathrm{d}\boldsymbol{x}\int p(\boldsymbol{x}')\,\mathrm{d}\boldsymbol{x}' - \int \pi_+ p(\boldsymbol{x})\ell(g(\boldsymbol{x}),+1)\,\mathrm{d}\boldsymbol{x}\int p_+(\boldsymbol{x}')\,\mathrm{d}\boldsymbol{x}' \\
&\quad + \int \pi_+ p_+(\boldsymbol{x})\ell(g(\boldsymbol{x}),+1)\,\mathrm{d}\boldsymbol{x}\int p(\boldsymbol{x}')\,\mathrm{d}\boldsymbol{x}' \\
&= \int \pi_+ p(\boldsymbol{x})\ell(g(\boldsymbol{x}),+1)\,\mathrm{d}\boldsymbol{x} - \int \pi_+ p(\boldsymbol{x})\ell(g(\boldsymbol{x}),+1)\,\mathrm{d}\boldsymbol{x} + \int \pi_+ p_+(\boldsymbol{x})\ell(g(\boldsymbol{x}),+1)\,\mathrm{d}\boldsymbol{x} \\
&= \int \pi_+ p_+(\boldsymbol{x})\ell(g(\boldsymbol{x}),+1)\,\mathrm{d}\boldsymbol{x} \\
&= \pi_+ \mathbb{E}_{p_+(\boldsymbol{x})}[\ell(g(\boldsymbol{x}),+1)].
\end{aligned}
$$

After that, the proof of Eq. (19) is given:

$$
\begin{aligned}
&\mathbb{E}_{p(\boldsymbol{x},\boldsymbol{x}')}[(\pi_- + c(\boldsymbol{x},\boldsymbol{x}'))\ell(g(\boldsymbol{x}),-1)] \\
&= \int\int \frac{\pi_- p(\boldsymbol{x})p(\boldsymbol{x}') + \pi_- p_-(\boldsymbol{x})p(\boldsymbol{x}') - \pi_- p(\boldsymbol{x})p_-(\boldsymbol{x}')}{p(\boldsymbol{x})p(\boldsymbol{x}')}\ell(g(\boldsymbol{x}),-1)p(\boldsymbol{x},\boldsymbol{x}')\,\mathrm{d}\boldsymbol{x}\,\mathrm{d}\boldsymbol{x}' \\
&= \int\int (\pi_- p(\boldsymbol{x})p(\boldsymbol{x}') + \pi_- p_-(\boldsymbol{x})p(\boldsymbol{x}') - \pi_- p(\boldsymbol{x})p_-(\boldsymbol{x}'))\ell(g(\boldsymbol{x}),-1)\,\mathrm{d}\boldsymbol{x}\,\mathrm{d}\boldsymbol{x}' \\
&= \int \pi_- p(\boldsymbol{x})\ell(g(\boldsymbol{x}),-1)\,\mathrm{d}\boldsymbol{x}\int p(\boldsymbol{x}')\,\mathrm{d}\boldsymbol{x}' + \int \pi_- p_-(\boldsymbol{x})\ell(g(\boldsymbol{x}),-1)\,\mathrm{d}\boldsymbol{x}\int p(\boldsymbol{x}')\,\mathrm{d}\boldsymbol{x}' \\
&\quad - \int \pi_- p(\boldsymbol{x})\ell(g(\boldsymbol{x}),-1)\,\mathrm{d}\boldsymbol{x}\int p_-(\boldsymbol{x}')\,\mathrm{d}\boldsymbol{x}' \\
&= \int \pi_- p(\boldsymbol{x})\ell(g(\boldsymbol{x}),-1)\,\mathrm{d}\boldsymbol{x} + \int \pi_- p_-(\boldsymbol{x})\ell(g(\boldsymbol{x}),-1)\,\mathrm{d}\boldsymbol{x} - \int \pi_- p(\boldsymbol{x})\ell(g(\boldsymbol{x}),-1)\,\mathrm{d}\boldsymbol{x} \\
&= \int \pi_- p_-(\boldsymbol{x})\ell(g(\boldsymbol{x}),-1)\,\mathrm{d}\boldsymbol{x} \\
&= \pi_- \mathbb{E}_{p_-(\boldsymbol{x})}[\ell(g(\boldsymbol{x}),-1)].
\end{aligned}
$$

It can be noticed that $c(\boldsymbol{x},\boldsymbol{x}') = -c(\boldsymbol{x}',\boldsymbol{x})$ and $p(\boldsymbol{x},\boldsymbol{x}') = p(\boldsymbol{x}',\boldsymbol{x})$. Therefore, it can be deduced naturally that $\mathbb{E}_{p(\boldsymbol{x},\boldsymbol{x}')}[(\pi_+ - c(\boldsymbol{x},\boldsymbol{x}'))\ell(g(\boldsymbol{x}),+1)] = \mathbb{E}_{p(\boldsymbol{x}',\boldsymbol{x})}[(\pi_+ + c(\boldsymbol{x}',\boldsymbol{x}))\ell(g(\boldsymbol{x}),+1)]$. Because $\boldsymbol{x}$ and $\boldsymbol{x}'$ are symmetric, we can swap them and deduce Eq. (20). Eq. (21) can be deduced in the same manner, which concludes the proof. $\square$

Based on Lemma 3, the proof of Theorem 1 is given.

*Proof of Theorem 1.* To begin with, it can be noticed that $\mathbb{E}_{p_+(\boldsymbol{x})}[\ell(g(\boldsymbol{x}),+1)] = \mathbb{E}_{p_+(\boldsymbol{x}')}[\ell(g(\boldsymbol{x}'),+1)]$ and $\mathbb{E}_{p_-(\boldsymbol{x})}[\ell(g(\boldsymbol{x}),-1)] = \mathbb{E}_{p_-(\boldsymbol{x}')}[\ell(g(\boldsymbol{x}'),-1)]$. Then, by summing up all the equations from Eq. (18) to Eq. (21), we can get the following equation:

$$
\begin{aligned}
&\mathbb{E}_{p(\boldsymbol{x},\boldsymbol{x}')}[\mathcal{L}_+(g(\boldsymbol{x}),g(\boldsymbol{x}')) + \mathcal{L}_-(g(\boldsymbol{x}),g(\boldsymbol{x}'))] \\
&= 2\pi_+ \mathbb{E}_{p_+(\boldsymbol{x})}[\ell(g(\boldsymbol{x}),+1)] + 2\pi_- \mathbb{E}_{p_-(\boldsymbol{x})}[\ell(g(\boldsymbol{x}),-1)]
\end{aligned}
$$

After dividing each side of the equation above by 2, we can obtain Theorem 1. $\square$

# B ANALYSIS ON VARIANCE OF RISK ESTIMATOR

## B.1 PROOF OF LEMMA 1

Based on Lemma 3, it can be observed that

$$
\begin{aligned}
\mathbb{E}_{p(\boldsymbol{x},\boldsymbol{x}')}[\mathcal{L}(\boldsymbol{x},\boldsymbol{x}')] &= \mathbb{E}_{p(\boldsymbol{x},\boldsymbol{x}')}[(\pi_+ - c(\boldsymbol{x},\boldsymbol{x}'))\ell(g(\boldsymbol{x}),+1) + (\pi_- - c(\boldsymbol{x},\boldsymbol{x}'))\ell(g(\boldsymbol{x}'),-1)] \\
&= \pi_+\mathbb{E}_{p_+(\boldsymbol{x})}[\ell(g(\boldsymbol{x}),+1)] + \pi_-\mathbb{E}_{p_-(\boldsymbol{x}')}[\ell(g(\boldsymbol{x}'),-1)] \\
&= \pi_+\mathbb{E}_{p_+(\boldsymbol{x})}[\ell(g(\boldsymbol{x}),+1)] + \pi_-\mathbb{E}_{p_-(\boldsymbol{x})}[\ell(g(\boldsymbol{x}),-1)] \\
&= R(g)
\end{aligned}
$$

and

$$
\begin{aligned}
\mathbb{E}_{p(\boldsymbol{x},\boldsymbol{x}')}[\mathcal{L}(\boldsymbol{x}',\boldsymbol{x})] &= \mathbb{E}_{p(\boldsymbol{x},\boldsymbol{x}')}[(\pi_+ + c(\boldsymbol{x},\boldsymbol{x}'))\ell(g(\boldsymbol{x}'),+1) + (\pi_- + c(\boldsymbol{x},\boldsymbol{x}'))\ell(g(\boldsymbol{x}),-1)] \\
&= \pi_-\mathbb{E}_{p_-(\boldsymbol{x})}[\ell(g(\boldsymbol{x}),-1)] + \pi_+\mathbb{E}_{p_+(\boldsymbol{x}')}[\ell(g(\boldsymbol{x}'),+1)] \\
&= \pi_-\mathbb{E}_{p_-(\boldsymbol{x})}[\ell(g(\boldsymbol{x}),-1)] + \pi_+\mathbb{E}_{p_+(\boldsymbol{x})}[\ell(g(\boldsymbol{x}),+1)] \\
&= R(g).
\end{aligned}
$$

Therefore, for an arbitrary weight $\alpha \in [0,1]$,

$$
\begin{aligned}
R(g) &= \alpha R(g) + (1-\alpha)R(g) \\
&= \alpha\mathbb{E}_{p(\boldsymbol{x},\boldsymbol{x}')}[\mathcal{L}(\boldsymbol{x},\boldsymbol{x}')] + (1-\alpha)\mathbb{E}_{p(\boldsymbol{x},\boldsymbol{x}')}[\mathcal{L}(\boldsymbol{x}',\boldsymbol{x})],
\end{aligned}
$$

which indicates that

$$
\frac{1}{n}\sum_{i=1}^{n}(\alpha\mathcal{L}(\boldsymbol{x}_i,\boldsymbol{x}_i') + (1-\alpha)\mathcal{L}(\boldsymbol{x}_i',\boldsymbol{x}_i))
$$

is also an unbiased risk estimator and concludes the proof. $\square$

## B.2 PROOF OF THEOREM 2

In this subsection, we show that Eq. (8) achieves the minimum variance of

$$
S(g;\alpha) = \frac{1}{n}\sum_{i=1}^{n}(\alpha\mathcal{L}(\boldsymbol{x}_i,\boldsymbol{x}_i') + (1-\alpha)\mathcal{L}(\boldsymbol{x}_i',\boldsymbol{x}_i))
$$

w.r.t. any $\alpha \in [0,1]$. To begin with, we introduce the following notations:

$$
\mu_1 \triangleq \mathbb{E}_{p(\boldsymbol{x},\boldsymbol{x}')}[(\frac{1}{n}\sum_{i=1}^{n}\mathcal{L}(\boldsymbol{x}_i,\boldsymbol{x}_i'))^2] = \mathbb{E}_{p(\boldsymbol{x},\boldsymbol{x}')}[(\frac{1}{n}\sum_{i=1}^{n}\mathcal{L}(\boldsymbol{x}_i',\boldsymbol{x}_i))^2],
$$

$$
\mu_2 \triangleq \mathbb{E}_{p(\boldsymbol{x},\boldsymbol{x}')}[\frac{1}{n^2}\sum_{i=1}^{n}\mathcal{L}(\boldsymbol{x}_i,\boldsymbol{x}_i')\sum_{i=1}^{n}\mathcal{L}(\boldsymbol{x}_i',\boldsymbol{x}_i)]. \tag{22}
$$

Furthermore, according to Lemma 1, we have

$$
\mathbb{E}_{p(\boldsymbol{x},\boldsymbol{x}')}[S(g;\alpha)] = R(g).
$$

Then, we provide the proof of Theorem 2 as follows.

*Proof of Theorem 2.*

$$
\begin{aligned}
\text{Var}(S(g;\alpha)) &= \mathbb{E}_{p(\boldsymbol{x},\boldsymbol{x}')}[(S(g;\alpha) - R(g))^2] \\
&= \mathbb{E}_{p(\boldsymbol{x},\boldsymbol{x}')}[S(g;\alpha)^2] - R(g)^2 \\
&= \alpha^2\mathbb{E}_{p(\boldsymbol{x},\boldsymbol{x}')}[(\frac{1}{n}\sum_{i=1}^{n}\mathcal{L}(\boldsymbol{x}_i,\boldsymbol{x}_i'))^2] + (1-\alpha)^2\mathbb{E}_{p(\boldsymbol{x},\boldsymbol{x}')}[(\frac{1}{n}\sum_{i=1}^{n}\mathcal{L}(\boldsymbol{x}_i',\boldsymbol{x}_i))^2] \\
&\quad + 2\alpha(1-\alpha)\mathbb{E}_{p(\boldsymbol{x},\boldsymbol{x}')}[\frac{1}{n^2}\sum_{i=1}^{n}\mathcal{L}(\boldsymbol{x}_i,\boldsymbol{x}_i')\sum_{i=1}^{n}\mathcal{L}(\boldsymbol{x}_i',\boldsymbol{x}_i)] - R(g)^2 \\
&= \mu_1\alpha^2 + \mu_1(1-\alpha)^2 + 2\mu_2\alpha(1-\alpha) - R(g)^2 \\
&= (2\mu_1 - 2\mu_2)(\alpha - \frac{1}{2})^2 + \frac{1}{2}(\mu_1 + \mu_2) - R(g)^2.
\end{aligned}
$$

Besides, it can be observed that

$$2\mu_1 - 2\mu_2 = \mathbb{E}_{p(\boldsymbol{x},\boldsymbol{x}')}[(\frac{1}{n}\sum_{i=1}^{n}(\mathcal{L}(\boldsymbol{x}_i,\boldsymbol{x}_i') - \mathcal{L}(\boldsymbol{x}_i',\boldsymbol{x}_i))^2] \geq 0.$$

Therefore, $\mathrm{Var}(S(g;\alpha))$ achieves the minimum value when $\alpha = 1/2$, which concludes the proof.
□

## C  PROOF OF THEOREM 3

To begin with, we give the definition of Rademacher complexity.

**Definition 2** (Rademacher complexity). *Let $\mathcal{X}_n = \{\boldsymbol{x}_1, \cdots \boldsymbol{x}_n\}$ denote $n$ i.i.d. random variables drawn from a probability distribution with density $p(\boldsymbol{x})$, $\mathcal{G} = \{g : \mathcal{X} \mapsto \mathbb{R}\}$ denote a class of measurable functions, and $\boldsymbol{\sigma} = (\sigma_1, \sigma_2, \cdots, \sigma_n)$ denote Rademacher variables taking values from $\{+1, -1\}$ uniformly. Then, the (expected) Rademacher complexity of $\mathcal{G}$ is defined as*

$$\mathfrak{R}_n(\mathcal{G}) = \mathbb{E}_{\mathcal{X}_n}\mathbb{E}_{\boldsymbol{\sigma}}\left[\sup_{g\in\mathcal{G}}\frac{1}{n}\sum_{i=1}^{n}\sigma_i g(\boldsymbol{x}_i)\right]. \tag{23}$$

Let $\mathcal{D}_n \overset{\text{i.i.d.}}{\sim} p(\boldsymbol{x},\boldsymbol{x}')$ denote $n$ pairs of ConfDiff data and $\mathcal{L}_{\mathrm{CD}}(g;\boldsymbol{x}_i,\boldsymbol{x}_i') = (\mathcal{L}(\boldsymbol{x},\boldsymbol{x}') + \mathcal{L}(\boldsymbol{x}',\boldsymbol{x}))/2$, then we introduce the following lemma.

**Lemma 4.**
$$\bar{\mathfrak{R}}_n(\mathcal{L}_{\mathrm{CD}} \circ \mathcal{G}) \leq 2L_\ell \mathfrak{R}_n(\mathcal{G}),$$

*where $\mathcal{L}_{\mathrm{CD}} \circ \mathcal{G} = \{\mathcal{L}_{\mathrm{CD}} \circ g | g \in \mathcal{G}\}$ and $\bar{\mathfrak{R}}_n(\cdot)$ is the Rademacher complexity over ConfDiff data pairs $\mathcal{D}_n$ of size $n$.*

*Proof.*

$$\begin{aligned}
\bar{\mathfrak{R}}_n(\mathcal{L}_{\mathrm{CD}} \circ \mathcal{G}) =& \mathbb{E}_{\mathcal{D}_n}\mathbb{E}_{\boldsymbol{\sigma}}[\sup_{g\in\mathcal{G}}\frac{1}{n}\sum_{i=1}^{n}\sigma_i\mathcal{L}_{\mathrm{CD}}(g;\boldsymbol{x}_i,\boldsymbol{x}_i')] \\
=& \mathbb{E}_{\mathcal{D}_n}\mathbb{E}_{\boldsymbol{\sigma}}[\sup_{g\in\mathcal{G}}\frac{1}{2n}\sum_{i=1}^{n}\sigma_i((\pi_+ - c_i)\ell(g(\boldsymbol{x}_i),+1) + (\pi_- - c_i)\ell(g(\boldsymbol{x}_i'),-1) \\
& + (\pi_+ + c_i)\ell(g(\boldsymbol{x}_i'),+1) + (\pi_- + c_i)\ell(g(\boldsymbol{x}_i),-1))].
\end{aligned}$$

Then, we can induce that

$$\begin{aligned}
& \|\nabla\mathcal{L}_{\mathrm{CD}}(g;\boldsymbol{x}_i,\boldsymbol{x}_i')\|_2 \\
=& \|\nabla(\frac{(\pi_+ - c_i)\ell(g(\boldsymbol{x}_i),+1) + (\pi_- - c_i)\ell(g(\boldsymbol{x}_i'),-1)}{2} \\
& + \frac{(\pi_+ + c_i)\ell(g(\boldsymbol{x}_i'),+1) + (\pi_- + c_i)\ell(g(\boldsymbol{x}_i),-1)}{2})\|_2 \\
\leq& \|\nabla(\frac{(\pi_+ - c_i)\ell(g(\boldsymbol{x}_i),+1)}{2})\|_2 + \|\nabla(\frac{(\pi_- - c_i)\ell(g(\boldsymbol{x}_i'),-1)}{2})\|_2 \\
& + \|\nabla(\frac{(\pi_+ + c_i)\ell(g(\boldsymbol{x}_i'),+1)}{2})\|_2 + \|\nabla(\frac{(\pi_- + c_i)\ell(g(\boldsymbol{x}_i),-1)}{2})\|_2 \\
\leq& \frac{|\pi_+ - c_i|L_\ell}{2} + \frac{|\pi_- - c_i|L_\ell}{2} + \frac{|\pi_+ + c_i|L_\ell}{2} + \frac{|\pi_- + c_i|L_\ell}{2}. \tag{24}
\end{aligned}$$

Suppose $\pi_+ \geq \pi_-$, the value of RHS of Eq. (24) can be determined as follows: when $c_i \in [-1, -\pi_+)$, the value is $-2c_iL_\ell$; when $c_i \in [-\pi_+, -\pi_-)$, the value is $(\pi_+ - c_i)L_\ell$; when $c_i \in [-\pi_-, \pi_-)$, the value is $L_\ell$; when $c_i \in [\pi_-, \pi_+)$, the value is $(\pi_+ + c_i)L_\ell$; when $c_i \in [\pi_+, 1]$, the value is $2c_iL_\ell$. To sum up, when $\pi_+ \geq \pi_-$, the value of RHS of Eq. (24) is less than $2L_\ell$.

When $\pi_+ \leq \pi_-$, we can deduce that the value of RHS of Eq. (24) is less than $2L_\ell$ in the same way. Therefore,

$$
\begin{aligned}
\bar{\mathfrak{R}}_n(\mathcal{L}_{\text{CD}} \circ \mathcal{G}) &\leq 2L_\ell \mathbb{E}_{\mathcal{D}_n} \mathbb{E}_{\boldsymbol{\sigma}} [\sup_{g \in \mathcal{G}} \frac{1}{n} \sum_{i=1}^{n} \sigma_i g(\boldsymbol{x}_i)] \\
&= 2L_\ell \mathbb{E}_{\mathcal{X}_n} \mathbb{E}_{\boldsymbol{\sigma}} [\sup_{g \in \mathcal{G}} \frac{1}{n} \sum_{i=1}^{n} \sigma_i g(\boldsymbol{x}_i)] \\
&= 2L_\ell \mathfrak{R}_n(\mathcal{G}),
\end{aligned}
$$

which concludes the proof. $\qquad\qquad\square$

After that, we introduce the following lemma.

**Lemma 5.** *The inequality below hold with probability at least $1 - \delta$:*

$$
\sup_{g \in \mathcal{G}} |R(g) - \widehat{R}_{\text{CD}}(g)| \leq 4L_\ell \mathfrak{R}_n(\mathcal{G}) + 2C_\ell \sqrt{\frac{\ln 2/\delta}{2n}}.
$$

*Proof.* To begin with, we introduce $\Phi = \sup_{g \in \mathcal{G}}(R(g) - \widehat{R}_{\text{CD}}(g))$ and $\bar{\Phi} = \sup_{g \in \mathcal{G}}(R(g) - \widehat{\bar{R}}_{\text{CD}}(g))$, where $\widehat{R}_{\text{CD}}(g)$ and $\widehat{\bar{R}}_{\text{CD}}(g)$ denote the empirical risk over two sets of training examples with exactly one different point $\{(\boldsymbol{x}_i, \boldsymbol{x}'_i), c_i\}$ and $\{(\bar{\boldsymbol{x}}_i, \bar{\boldsymbol{x}}'_i), c(\bar{\boldsymbol{x}}_i, \bar{\boldsymbol{x}}'_i)\}$ respectively. Then we have

$$
\begin{aligned}
\bar{\Phi} - \Phi &\leq \sup_{g \in \mathcal{G}}(\widehat{R}_{\text{CD}}(g) - \widehat{\bar{R}}_{\text{CD}}(g)) \\
&\leq \sup_{g \in \mathcal{G}} (\frac{\mathcal{L}_{\text{CD}}(g; \boldsymbol{x}_i, \boldsymbol{x}'_i) - \mathcal{L}_{\text{CD}}(g; \bar{\boldsymbol{x}}_i, \bar{\boldsymbol{x}}'_i)}{n}) \\
&\leq \frac{2C_\ell}{n}.
\end{aligned}
$$

Accordingly, $\Phi - \bar{\Phi}$ can be bounded in the same way. The following inequalities holds with probability at least $1 - \delta/2$ by applying McDiarmid's inequality:

$$
\sup_{g \in \mathcal{G}}(R(g) - \widehat{R}_{\text{CD}}(g)) \leq \mathbb{E}_{\mathcal{D}_n}[\sup_{g \in \mathcal{G}}(R(g) - \widehat{R}_{\text{CD}}(g))] + 2C_\ell \sqrt{\frac{\ln 2/\delta}{2n}},
$$

Furthermore, we can bound $\mathbb{E}_{\mathcal{D}_n}[\sup_{g \in \mathcal{G}}(R(g) - \widehat{R}_{\text{CD}}(g))]$ with Rademacher complexity. It is a routine work to show by symmetrization (Mohri et al., 2012) that

$$
\mathbb{E}_{\mathcal{D}_n}[\sup_{g \in \mathcal{G}}(R(g) - \widehat{R}_{\text{CD}}(g))] \leq 2\bar{\mathfrak{R}}_n(\mathcal{L}_{\text{CD}} \circ \mathcal{G}) \leq 4L_\ell \mathfrak{R}_n(\mathcal{G}),
$$

where the second inequality is from Lemma 4. Accordingly, $\sup_{g \in \mathcal{G}}(\widehat{R}_{\text{CD}}(g) - R(g))$ has the same bound. By using the union bound, the following inequality holds with probability at least $1 - \delta$:

$$
\sup_{g \in \mathcal{G}} |R(g) - \widehat{R}_{\text{CD}}(g)| \leq 4L_\ell \mathfrak{R}_n(\mathcal{G}) + 2C_\ell \sqrt{\frac{\ln 2/\delta}{2n}},
$$

which concludes the proof. $\qquad\qquad\square$

Finally, the proof of Theorem 3 is provided.

*Proof of Theorem 3.*

$$
\begin{aligned}
R(\widehat{g}_{\text{CD}}) - R(g^*) &= (R(\widehat{g}_{\text{CD}}) - \widehat{R}_{\text{CD}}(\widehat{g}_{\text{CD}})) + (\widehat{R}_{\text{CD}}(\widehat{g}_{\text{CD}}) - \widehat{R}_{\text{CD}}(g^*)) + (\widehat{R}_{\text{CD}}(g^*) - R(g^*)) \\
&\leq (R(\widehat{g}_{\text{CD}}) - \widehat{R}_{\text{CD}}(\widehat{g}_{\text{CD}})) + (\widehat{R}_{\text{CD}}(g^*) - R(g^*)) \\
&\leq |R(\widehat{g}_{\text{CD}}) - \widehat{R}_{\text{CD}}(\widehat{g}_{\text{CD}})| + \left|\widehat{R}_{\text{CD}}(g^*) - R(g^*)\right| \\
&\leq 2\sup_{g \in \mathcal{G}} |R(g) - \widehat{R}_{\text{CD}}(g)| \\
&\leq 8L_\ell \mathfrak{R}_n(\mathcal{G}) + 4C_\ell \sqrt{\frac{\ln 2/\delta}{2n}}.
\end{aligned}
$$

The first inequality is derived because $\widehat{g}_{\mathrm{CD}}$ is the minimizer of $\widehat{R}_{\mathrm{CD}}(g)$. The last inequality is derived according to Lemma 5, which concludes the proof. $\qquad\square$

## D  PROOF OF THEOREM 4

To begin with, we provide the following inequality:

$$
\begin{aligned}
&\sup_{g\in\mathcal{G}}|\bar{R}_{\mathrm{CD}}(g)-\widehat{R}_{\mathrm{CD}}(g)|\\
=&\frac{1}{2n}|\sum_{i=1}^{n}((\bar{\pi}_{+}-\pi_{+}+c_{i}-\bar{c}_{i})\ell(g(\boldsymbol{x}_{i}),+1)+(\bar{\pi}_{-}-\pi_{-}+c_{i}-\bar{c}_{i})\ell(g(\boldsymbol{x}'_{i}),-1)\\
&+(\bar{\pi}_{+}-\pi_{+}+\bar{c}_{i}-c_{i})\ell(g(\boldsymbol{x}'_{i}),+1)+(\bar{\pi}_{-}-\pi_{-}+\bar{c}_{i}-c_{i})\ell(g(\boldsymbol{x}_{i}),-1))|\\
\leq&\frac{1}{2n}\sum_{i=1}^{n}(|(\bar{\pi}_{+}-\pi_{+}+c_{i}-\bar{c}_{i})\ell(g(\boldsymbol{x}_{i}),+1)|+|(\bar{\pi}_{-}-\pi_{-}+c_{i}-\bar{c}_{i})\ell(g(\boldsymbol{x}'_{i}),-1)|\\
&+|(\bar{\pi}_{+}-\pi_{+}+\bar{c}_{i}-c_{i})\ell(g(\boldsymbol{x}'_{i}),+1)|+|(\bar{\pi}_{-}-\pi_{-}+\bar{c}_{i}-c_{i})\ell(g(\boldsymbol{x}_{i}),-1)|)\\
=&\frac{1}{2n}\sum_{i=1}^{n}(|\bar{\pi}_{+}-\pi_{+}+c_{i}-\bar{c}_{i}|\ell(g(\boldsymbol{x}_{i}),+1)+|\bar{\pi}_{-}-\pi_{-}+c_{i}-\bar{c}_{i}|\ell(g(\boldsymbol{x}'_{i}),-1)\\
&+|\bar{\pi}_{+}-\pi_{+}+\bar{c}_{i}-c_{i}|\ell(g(\boldsymbol{x}'_{i}),+1)+|\bar{\pi}_{-}-\pi_{-}+\bar{c}_{i}-c_{i}|\ell(g(\boldsymbol{x}_{i}),-1))\\
\leq&\frac{1}{2n}\sum_{i=1}^{n}((|\bar{\pi}_{+}-\pi_{+}|+|c_{i}-\bar{c}_{i}|)\ell(g(\boldsymbol{x}_{i}),+1)+(|\bar{\pi}_{-}-\pi_{-}|+|c_{i}-\bar{c}_{i}|)\ell(g(\boldsymbol{x}'_{i}),-1)\\
&+(|\bar{\pi}_{+}-\pi_{+}|+|\bar{c}_{i}-c_{i}|)\ell(g(\boldsymbol{x}'_{i}),+1)+(|\bar{\pi}_{-}-\pi_{-}|+|\bar{c}_{i}-c_{i}|)\ell(g(\boldsymbol{x}_{i}),-1))\\
=&\frac{1}{2n}\sum_{i=1}^{n}((|\bar{\pi}_{+}-\pi_{+}|+|c_{i}-\bar{c}_{i}|)\ell(g(\boldsymbol{x}_{i}),+1)+(|\pi_{+}-\bar{\pi}_{+}|+|c_{i}-\bar{c}_{i}|)\ell(g(\boldsymbol{x}'_{i}),-1)\\
&+(|\bar{\pi}_{+}-\pi_{+}|+|\bar{c}_{i}-c_{i}|)\ell(g(\boldsymbol{x}'_{i}),+1)+(|\pi_{+}-\bar{\pi}_{+}|+|\bar{c}_{i}-c_{i}|)\ell(g(\boldsymbol{x}_{i}),-1))\\
\leq&\frac{2C_{\ell}\sum_{i=1}^{n}|\bar{c}_{i}-c_{i}|}{n}+2C_{\ell}|\bar{\pi}_{+}-\pi_{+}|.
\end{aligned}
$$

Then, we deduce the following inequality:

$$
\begin{aligned}
R(\bar{g}_{\mathrm{CD}})-R(g^{*})=&(R(\bar{g}_{\mathrm{CD}})-\widehat{R}_{\mathrm{CD}}(\bar{g}_{\mathrm{CD}}))+(\widehat{R}_{\mathrm{CD}}(\bar{g}_{\mathrm{CD}})-\bar{R}_{\mathrm{CD}}(\bar{g}_{\mathrm{CD}}))+(\bar{R}_{\mathrm{CD}}(\bar{g}_{\mathrm{CD}})-\bar{R}_{\mathrm{CD}}(\widehat{g}_{\mathrm{CD}}))\\
&+(\bar{R}_{\mathrm{CD}}(\widehat{g}_{\mathrm{CD}})-\widehat{R}_{\mathrm{CD}}(\widehat{g}_{\mathrm{CD}}))+(\widehat{R}_{\mathrm{CD}}(\widehat{g}_{\mathrm{CD}})-R(\widehat{g}_{\mathrm{CD}}))+(R(\widehat{g}_{\mathrm{CD}})-R(g^{*}))\\
\leq&2\sup_{g\in\mathcal{G}}|R(g)-\widehat{R}_{\mathrm{CD}}(g)|+2\sup_{g\in\mathcal{G}}|\bar{R}_{\mathrm{CD}}(g)-\widehat{R}_{\mathrm{CD}}(g)|+(R(\widehat{g}_{\mathrm{CD}})-R(g^{*}))\\
\leq&4\sup_{g\in\mathcal{G}}|R(g)-\widehat{R}_{\mathrm{CD}}(g)|+2\sup_{g\in\mathcal{G}}|\bar{R}_{\mathrm{CD}}(g)-\widehat{R}_{\mathrm{CD}}(g)|\\
\leq&16L_{\ell}\mathfrak{R}_{n}(\mathcal{G})+8C_{\ell}\sqrt{\frac{\ln 2/\delta}{2n}}+\frac{4C_{\ell}\sum_{i=1}^{n}|\bar{c}_{i}-c_{i}|}{n}+4C_{\ell}|\bar{\pi}_{+}-\pi_{+}|.
\end{aligned}
$$

The first inequality is derived because $\bar{g}_{\mathrm{CD}}$ is the minimizer of $\bar{R}(g)$. The second and third inequality are derived according to the proof of Theorem 3 and Lemma 5 respectively. $\qquad\square$

## E  PROOF OF THEOREM 5

To begin with, let $\mathfrak{D}_{n}^{+}(g)=\{\mathcal{D}_{n}|\widehat{A}(g)\geq 0\cap\widehat{B}(g)\geq 0\cap\widehat{C}(g)\geq 0\cap\widehat{D}(g)\geq 0\}$ and $\mathfrak{D}_{n}^{-}(g)=\{\mathcal{D}_{n}|\widehat{A}(g)\leq 0\cup\widehat{B}(g)\leq 0\cup\widehat{C}(g)\leq 0\cup\widehat{D}(g)\leq 0\}$. Before giving the proof of Theorem 5, we give the following lemma based on the assumptions in section 3.

**Lemma 6.** *The probability measure of $\mathfrak{D}_{n}^{-}(g)$ can be bounded as follows:*

$$
\mathbb{P}(\mathfrak{D}_{n}^{-}(g))\leq\exp\left(\frac{-2a^{2}n}{C_{\ell}^{2}}\right)+\exp\left(\frac{-2b^{2}n}{C_{\ell}^{2}}\right)+\exp\left(\frac{-2c^{2}n}{C_{\ell}^{2}}\right)+\exp\left(\frac{-2d^{2}n}{C_{\ell}^{2}}\right). \tag{25}
$$

*Proof.* It can be observed that

$$
\begin{aligned}
p(\mathcal{D}_n) &= p(\boldsymbol{x}_1, \boldsymbol{x}_1') \cdots p(\boldsymbol{x}_n, \boldsymbol{x}_n') \\
&= p(\boldsymbol{x}_1) \cdots p(\boldsymbol{x}_n') p(\boldsymbol{x}_1) \cdots p(\boldsymbol{x}_n').
\end{aligned}
$$

Therefore, the probability measure $\mathbb{P}(\mathfrak{D}_n^-(g))$ can be defined as follows:

$$
\begin{aligned}
\mathbb{P}(\mathfrak{D}_n^-(g)) &= \int_{\mathcal{D}_n \in \mathfrak{D}_n^-(g)} p(\mathcal{D}_n) \, \mathrm{d}\mathcal{D}_n \\
&= \int_{\mathcal{D}_n \in \mathfrak{D}_n^-(g)} p(\mathcal{D}_n) \, \mathrm{d}\boldsymbol{x}_1 \cdots \mathrm{d}\boldsymbol{x}_n \, \mathrm{d}\boldsymbol{x}_1' \cdots \mathrm{d}\boldsymbol{x}_n'.
\end{aligned}
$$

When exactly one ConfDiff data pair in $S_n$ is replaced, the change of $\widehat{A}(g), \widehat{B}(g), \widehat{C}(g)$ and $\widehat{D}(g)$ will be no more than $C_\ell/n$. By applying McDiarmid's inequality, we can obtain the following inequalities:

$$
\mathbb{P}(\mathbb{E}[\widehat{A}(g)] - \widehat{A}(g) \geq a) \leq \exp\big(\frac{-2a^2 n}{C_\ell^2}\big),
$$

$$
\mathbb{P}(\mathbb{E}[\widehat{B}(g)] - \widehat{B}(g) \geq b) \leq \exp\big(\frac{-2b^2 n}{C_\ell^2}\big),
$$

$$
\mathbb{P}(\mathbb{E}[\widehat{C}(g)] - \widehat{C}(g) \geq c) \leq \exp\big(\frac{-2c^2 n}{C_\ell^2}\big),
$$

$$
\mathbb{P}(\mathbb{E}[\widehat{D}(g)] - \widehat{D}(g) \geq d) \leq \exp\big(\frac{-2d^2 n}{C_\ell^2}\big).
$$

Furthermore,

$$
\begin{aligned}
\mathbb{P}(\mathfrak{D}_n^-(g) \leq &\, \mathbb{P}(\widehat{A}(g) \leq 0) + \mathbb{P}(\widehat{B}(g) \leq 0) + \mathbb{P}(\widehat{C}(g) \leq 0) + \mathbb{P}(\widehat{D}(g) \leq 0) \\
\leq &\, \mathbb{P}(\widehat{A}(g) \leq \mathbb{E}[\widehat{A}(g)] - a) + \mathbb{P}(\widehat{B}(g) \leq \mathbb{E}[\widehat{B}(g)] - b) \\
&+ \mathbb{P}(\widehat{C}(g) \leq \mathbb{E}[\widehat{C}(g)] - c) + \mathbb{P}(\widehat{D}(g) \leq \mathbb{E}[\widehat{D}(g)] - d) \\
\leq &\, \mathbb{P}(\mathbb{E}[\widehat{A}(g)] - \widehat{A}(g) \geq a) + \mathbb{P}(\mathbb{E}[\widehat{B}(g)] - \widehat{B}(g) \geq b) \\
&+ \mathbb{P}(\mathbb{E}[\widehat{C}(g)] - \widehat{C}(g) \geq c) + \mathbb{P}(\mathbb{E}[\widehat{D}(g)] - \widehat{D}(g) \geq d) \\
\leq &\, \exp\big(\frac{-2a^2 n}{C_\ell^2}\big) + \exp\big(\frac{-2b^2 n}{C_\ell^2}\big) + \exp\big(\frac{-2c^2 n}{C_\ell^2}\big) + \exp\big(\frac{-2d^2 n}{C_\ell^2}\big),
\end{aligned}
$$

which concludes the proof. $\qquad\square$

Then, the proof of Theorem 5 is given.

*Proof of Theorem 5.* To begin with, we prove the first inequality in Theorem 5.

$$
\begin{aligned}
&\mathbb{E}[\widetilde{R}_{\mathrm{CD}}(g)] - R(g) \\
=&\mathbb{E}[\widetilde{R}_{\mathrm{CD}}(g) - \widehat{R}_{\mathrm{CD}}(g)] \\
=&\int_{\mathcal{D}_n \in \mathfrak{D}_n^+(g)} (\widetilde{R}_{\mathrm{CD}}(g) - \widehat{R}_{\mathrm{CD}}(g)) p(\mathcal{D}_n) \, \mathrm{d}\mathcal{D}_n \\
&+ \int_{\mathcal{D}_n \in \mathfrak{D}_n^-(g)} (\widetilde{R}_{\mathrm{CD}}(g) - \widehat{R}_{\mathrm{CD}}(g)) p(\mathcal{D}_n) \, \mathrm{d}\mathcal{D}_n \\
=&\int_{\mathcal{D}_n \in \mathfrak{D}_n^-(g)} (\widetilde{R}_{\mathrm{CD}}(g) - \widehat{R}_{\mathrm{CD}}(g)) p(\mathcal{D}_n) \, \mathrm{d}\mathcal{D}_n \geq 0,
\end{aligned}
$$

where the last inequality is derived because $\widetilde{R}_{\mathrm{CD}}(g)$ is an upper bound of $\widehat{R}_{\mathrm{CD}}(g)$. Furthermore,

$$
\begin{aligned}
&\mathbb{E}[\widetilde{R}_{\mathrm{CD}}(g)] - R(g)\\
&= \int_{\mathcal{D}_n \in \mathfrak{D}_n^-(g)} (\widetilde{R}_{\mathrm{CD}}(g) - \widehat{R}_{\mathrm{CD}}(g)) p(\mathcal{D}_n)\, \mathrm{d}\mathcal{D}_n\\
&\leq \sup_{\mathcal{D}_n \in \mathfrak{D}_n^-(g)} (\widetilde{R}_{\mathrm{CD}}(g) - \widehat{R}_{\mathrm{CD}}(g)) \int_{\mathcal{D}_n \in \mathfrak{D}_n^-(g)} p(\mathcal{D}_n)\, \mathrm{d}\mathcal{D}_n\\
&= \sup_{\mathcal{D}_n \in \mathfrak{D}_n^-(g)} (\widetilde{R}_{\mathrm{CD}}(g) - \widehat{R}_{\mathrm{CD}}(g)) \mathbb{P}(\mathfrak{D}_n^-(g))\\
&= \sup_{\mathcal{D}_n \in \mathfrak{D}_n^-(g)} (f(\widehat{A}(g)) + f(\widehat{B}(g)) + f(\widehat{C}(g)) + f(\widehat{D}(g))\\
&\qquad - \widehat{A}(g) - \widehat{B}(g) - \widehat{C}(g) - \widehat{D}(g)) \mathbb{P}(\mathfrak{D}_n^-(g))\\
&\leq \sup_{\mathcal{D}_n \in \mathfrak{D}_n^-(g)} (L_f|\widehat{A}(g)| + L_f|\widehat{B}(g)| + L_f|\widehat{C}(g)| + L_f|\widehat{D}(g)|\\
&\qquad + |\widehat{A}(g)| + |\widehat{B}(g)| + |\widehat{C}(g)| + |\widehat{D}(g)|) \mathbb{P}(\mathfrak{D}_n^-(g)\\
&= \sup_{\mathcal{D}_n \in \mathfrak{D}_n^-(g)} \frac{L_f+1}{2n} (|\sum_{i=1}^n (\pi_+ - c_i)\ell(g(\boldsymbol{x}_i), +1)| + |\sum_{i=1}^n (\pi_- - c_i)\ell(g(\boldsymbol{x}_i'), -1)|\\
&\qquad + |\sum_{i=1}^n (\pi_+ + c_i)\ell(g(\boldsymbol{x}_i'), +1)| + |\sum_{i=1}^n (\pi_- + c_i)\ell(g(\boldsymbol{x}_i), -1)|) \mathbb{P}(\mathfrak{D}_n^-(g))\\
&\leq \sup_{\mathcal{D}_n \in \mathfrak{D}_n^-(g)} \frac{L_f+1}{2n} (\sum_{i=1}^n |(\pi_+ - c_i)\ell(g(\boldsymbol{x}_i), +1)| + \sum_{i=1}^n |(\pi_- - c_i)\ell(g(\boldsymbol{x}_i'), -1)|\\
&\qquad + \sum_{i=1}^n |(\pi_+ + c_i)\ell(g(\boldsymbol{x}_i'), +1)| + \sum_{i=1}^n |(\pi_- + c_i)\ell(g(\boldsymbol{x}_i), -1)|) \mathbb{P}(\mathfrak{D}_n^-(g))\\
&= \sup_{\mathcal{D}_n \in \mathfrak{D}_n^-(g)} \frac{L_f+1}{2n} \sum_{i=1}^n (|(\pi_+ - c_i)\ell(g(\boldsymbol{x}_i), +1)| + |(\pi_- - c_i)\ell(g(\boldsymbol{x}_i'), -1)|\\
&\qquad + |(\pi_+ + c_i)\ell(g(\boldsymbol{x}_i'), +1)| + |(\pi_- + c_i)\ell(g(\boldsymbol{x}_i), -1)|) \mathbb{P}(\mathfrak{D}_n^-(g))\\
&\leq \sup_{\mathcal{D}_n \in \mathfrak{D}_n^-(g)} \frac{(L_f+1)C_\ell}{2n} \sum_{i=1}^n (|\pi_+ - c_i| + |\pi_- - c_i| + |\pi_+ + c_i| + |\pi_- + c_i|) \mathbb{P}(\mathfrak{D}_n^-(g)).
\end{aligned}
$$

Similar to the proof of Theorem 3, we can obtain

$$
|\pi_+ - c_i| + |\pi_- - c_i| + |\pi_+ + c_i| + |\pi_- + c_i| \leq 4.
$$

Therefore, we have

$$
\mathbb{E}[\widetilde{R}_{\mathrm{CD}}(g)] - R(g) \leq 2(L_f+1)C_\ell\Delta,
$$

which concludes the proof of the first inequality in Theorem 5. Before giving the proof of the second inequality, we give the upper bound of $|\widetilde{R}_{\mathrm{CD}}(g) - \mathbb{E}[\widetilde{R}_{\mathrm{CD}}(g)]|$. When exactly one ConfDiff data pair in $\mathcal{D}_n$ is replaced, the change of $\widetilde{R}_{\mathrm{CD}}(g)$ is no more than $2C_\ell L_f/n$. By applying McDiarmid's inequality, we have the following inequalities with probability at least $1 - \delta/2$:

$$
\widetilde{R}_{\mathrm{CD}}(g) - \mathbb{E}[\widetilde{R}_{\mathrm{CD}}(g)] \leq 2C_\ell L_f \sqrt{\frac{\ln 2/\delta}{2n}},
$$

$$
\mathbb{E}[\widetilde{R}_{\mathrm{CD}}(g)] - \widetilde{R}_{\mathrm{CD}}(g) \leq 2C_\ell L_f \sqrt{\frac{\ln 2/\delta}{2n}}.
$$

Therefore, with probability at least $1 - \delta$, we have

$$
|\widetilde{R}_{\mathrm{CD}}(g) - \mathbb{E}[\widetilde{R}_{\mathrm{CD}}(g)]| \leq 2C_\ell L_f \sqrt{\frac{\ln 2/\delta}{2n}}.
$$

Table 3: Characteristics of experimental data sets.

| Data Set | # Train | # Test | # Features | # Class Labels | Model |
|---|---|---|---|---|---|
| **MNIST** | 60,000 | 10,000 | 784 | 10 | MLP |
| **Kuzushiji** | 60,000 | 10,000 | 784 | 10 | MLP |
| **Fashion** | 60,000 | 10,000 | 784 | 10 | MLP |
| **CIFAR-10** | 50,000 | 10,000 | 3,072 | 10 | ResNet-34 |
| **Optdigits** | 4,495 | 1,125 | 62 | 10 | MLP |
| **USPS** | 7,437 | 1,861 | 256 | 10 | MLP |
| **Pendigits** | 8,793 | 2,199 | 16 | 10 | MLP |
| **Letter** | 16,000 | 4,000 | 16 | 26 | MLP |

Finally, we have

$$
\begin{aligned}
|\widetilde{R}_{\mathrm{CD}}(g) - R(g)| &= |\widetilde{R}_{\mathrm{CD}}(g) - \mathbb{E}[\widetilde{R}_{\mathrm{CD}}(g)] + \mathbb{E}[\widetilde{R}_{\mathrm{CD}}(g)] - R(g)| \\
&\leq |\widetilde{R}_{\mathrm{CD}}(g) - \mathbb{E}[\widetilde{R}_{\mathrm{CD}}(g)]| + |\mathbb{E}[\widetilde{R}_{\mathrm{CD}}(g)] - R(g)| \\
&= |\widetilde{R}_{\mathrm{CD}}(g) - \mathbb{E}[\widetilde{R}_{\mathrm{CD}}(g)]| + \mathbb{E}[\widetilde{R}_{\mathrm{CD}}(g)] - R(g) \\
&\leq 2C_\ell L_f \sqrt{\frac{\ln 2/\delta}{2n}} + 2(L_f + 1)C_\ell \Delta,
\end{aligned}
\tag{26}
$$

with probability at least $1 - \delta$, which concludes the proof. $\qquad\square$

## F  PROOF OF THEOREM 6

With probability at least $1 - \delta$, we have

$$
\begin{aligned}
R(\widetilde{g}_{\mathrm{CD}}) - R(g^*) =& (R(\widetilde{g}_{\mathrm{CD}}) - \widetilde{R}_{\mathrm{CD}}(\widetilde{g}_{\mathrm{CD}})) + (\widetilde{R}_{\mathrm{CD}}(\widetilde{g}_{\mathrm{CD}}) - \widetilde{R}_{\mathrm{CD}}(\widehat{g}_{\mathrm{CD}})) \\
&+ (\widetilde{R}_{\mathrm{CD}}(\widehat{g}_{\mathrm{CD}}) - R(\widehat{g}_{\mathrm{CD}})) + (R(\widehat{g}_{\mathrm{CD}}) - R(g^*)) \\
\leq& |R(\widetilde{g}_{\mathrm{CD}}) - \widetilde{R}_{\mathrm{CD}}(\widetilde{g}_{\mathrm{CD}})| + |\widetilde{R}_{\mathrm{CD}}(\widehat{g}_{\mathrm{CD}}) - R(\widehat{g}_{\mathrm{CD}})| + (R(\widehat{g}_{\mathrm{CD}}) - R(g^*)) \\
\leq& 4C_\ell(L_f + 1)\sqrt{\frac{\ln 2/\delta}{2n}} + 4(L_f + 1)C_\ell \Delta + 8L_\ell \mathfrak{R}_n(\mathcal{G}).
\end{aligned}
$$

The first inequality is derived because $\widetilde{g}_{\mathrm{CD}}$ is the minimizer of $\widetilde{R}_{\mathrm{CD}}(g)$. The second inequality is derived from Theorem 5 and Theorem 3. The proof is completed. $\qquad\square$

## G  ADDITIONAL INFORMATION ON EXPERIMENTS

In this section, the details of experimental data sets and hyperparameters are provided.

### G.1  DETAILS OF EXPERIMENTAL DATA SETS

The detailed statistics and corresponding model architectures are summarized in Table 3 while the basic information, sources and data split details are elaborated in this subsection.

For the four benchmark data sets,

- MNIST (LeCun et al., 1998): It is a grayscale handwritten digits recognition data set. It is composed of 60,000 training examples and 10,000 test examples. The original feature dimension is 28*28, and the label space is 0-9. The even digits are regarded as the positive class while the odd digits are regarded as the negative class. We sampled 15,000 unlabeled data pairs as training data. The data set can be downloaded from http://yann.lecun.com/exdb/mnist/.
- Kuzushiji-MNIST (Clanuwat et al., 2018): It is a grayscale Japanese character recognition data set. It is composed of 60,000 training examples and 10,000 test examples. The original feature dimension is 28*28, and the label space is {'o', 'su','na', 'ma', 're', 'ki','tsu','ha', 'ya','wo'}.

The positive class is composed of 'o', 'su','na', 'ma', and 're' while the negative class is composed of 'ki','tsu','ha', 'ya', and 'wo'. We sampled 15,000 unlabeled data pairs as training data. The data set can be downloaded from `https://github.com/rois-codh/kmnist`.

- Fashion-MNIST (Xiao et al., 2017): It is a grayscale fashion item recognition data set. It is composed of 60,000 training examples and 10,000 test examples. The original feature dimension is 28*28, and the label space is {'T-shirt', 'trouser', 'pullover', 'dress', 'sandal', 'coat', 'shirt', 'sneaker', 'bag', 'ankle boot'}. The positive class is composed of 'T-shirt', 'pullover', 'coat', 'shirt', and 'bag' while the negative class is composed of 'trouser', 'dress', 'sandal', 'sneaker', and 'ankle boot'. We sampled 15,000 unlabeled data pairs as training data. The data set can be downloaded from `https://github.com/zalandoresearch/fashion-mnist`.
- CIFAR-10 (Krizhevsky & Hinton, 2009): It is a colorful object recognition data set. It is composed of 50,000 training examples and 10,000 test examples. The original feature dimension is 32*32*3, and the label space is {'airplane', 'bird', 'automobile', 'cat', 'deer', 'dog', 'frog', 'horse', 'ship', 'truck'}. The positive class is composed of 'bird', 'deer', 'dog', 'frog', 'cat', and 'horse' while the negative class is composed of 'airplane', 'automobile', 'ship', and 'truck'. We sampled 10,000 unlabeled data pairs as training data. The data set can be downloaded from `https://www.cs.toronto.edu/~kriz/cifar.html`.

For the four UCI data sets, they can be downloaded from Dua & Graff (2017).

- Optdigits, USPS, Pendigits (Dua & Graff, 2017): They are handwritten digit recognition data set. The train-test split can be found in Table 3. The feature dimensions are 62, 256, and 16 respectively and the label space is 0-9. The even digits are regarded as the positive class while the odd digits are regarded as the negative class. We sampled 1,200, 2,000, and 2,500 unlabeled data pairs for training respectively.
- Letter (Dua & Graff, 2017): It is a letter recognition data set. It is composed of 16,000 training examples and 4,000 test examples. The feature dimension is 16 and the label space is the 26 capital letters in the English alphabet. The positive class is composed of the top 13 letters while the negative class is composed of the latter 13 letters. We sampled 4,000 unlabeled data pairs for training.

### G.2 DETAILS OF HYPERPARAMETERS

All the methods were implemented in Pytorch (Paszke et al., 2019). We used the Adam optimizer (Kingma & Ba, 2015). To ensure fair comparisons, We set the same hyperparameter values for all the comparing approaches.

For MNIST, Kuzushiji-MNIST and Fashion-MNIST, the learning rate was set to 1e-3 and the weight decay was set to 1e-5. The batch size was set to 256 data pairs. For training the probabilistic classifier to generate confidence, the batch size was set to 256 and the epoch number was set to 10.

For CIFAR10, the learning rate was set to 5e-4 and the weight decay was set to 1e-5. The batch size was set to 128 data pairs. For training the probabilistic classifier to generate confidence, the batch size was set to 128 and the epoch number was set to 10.

For all the UCI data sets, the learning rate was set to 1e-3 and the weight decay was set to 1e-5. The batch size was set to 128 data pairs. For training the probabilistic classifier to generate confidence, the batch size was set to 128 and the epoch number was set to 10.

The learning rate and weight decay for training the probabilistic classifier were the same as the setting for each data set correspondingly.

## H  MORE EXPERIMENTAL RESULTS WITH FEWER TRAINING DATA

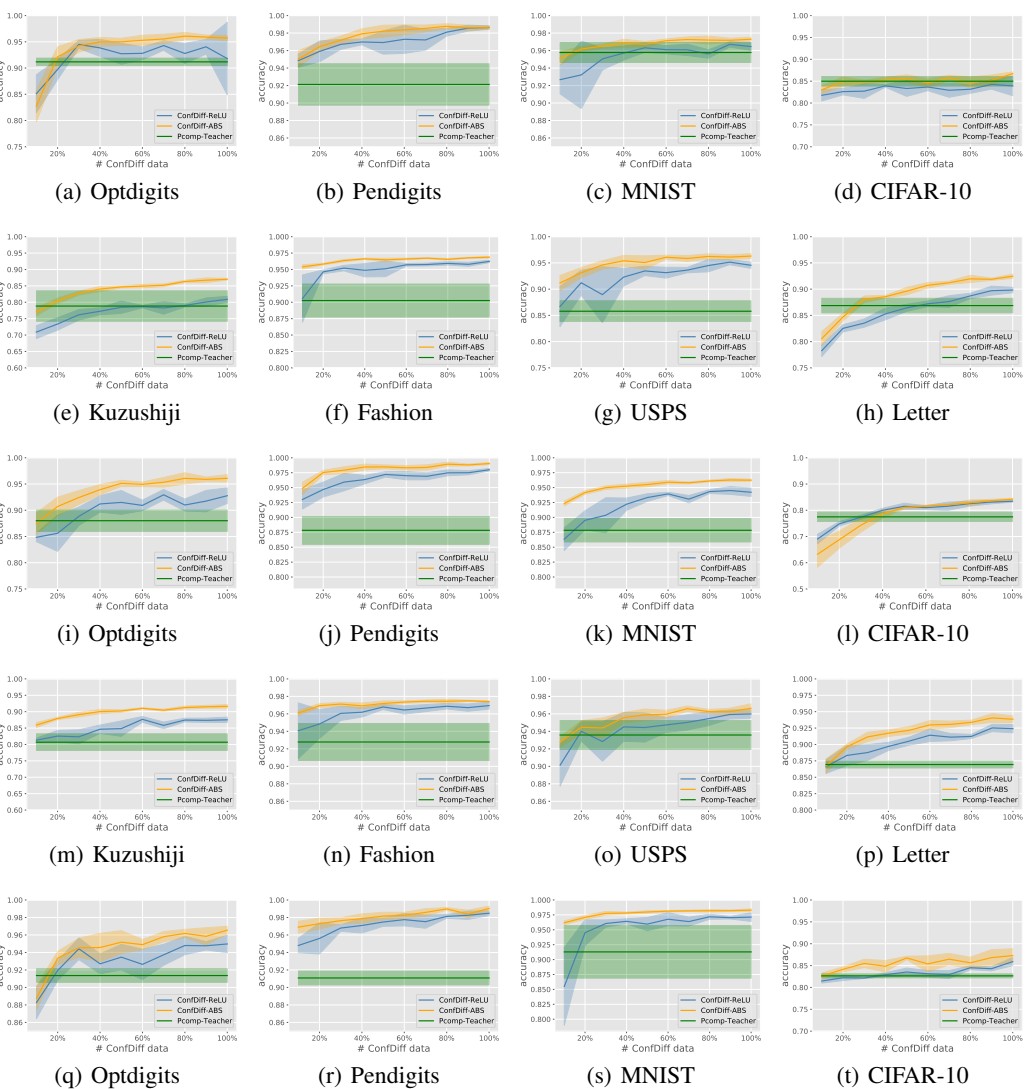

Figure 3: Classification performance of ConfDiff-ReLU and ConfDiff-ABS given a fraction of training data as well as Pcomp-Teacher given 100% of training data with different prior settings ($\pi_+ = 0.2$ for the first row, $\pi_+ = 0.5$ for the second and the third row, and $\pi_+ = 0.8$ for the fourth and the fifth row).

