# OpenReview forum: "Pairwise Confidence Difference on Unlabeled Data is Sufficient for Binary Classification"
_ICLR.cc/2023/Conference — Submitted to ICLR 2023_

### Official Review · Reviewer_dSMd · 2022-10-24

**Confidence:** 4
**Correctness:** 4
**Technical Novelty And Significance:** 3
**Empirical Novelty And Significance:** 3
**Recommendation:** 6

**Clarity, Quality, Novelty And Reproducibility:**

The paper is well-organized and well-written. I enjoyed reading it. Overall it is novel at large. The reproducibility is good.

**Details Of Ethics Concerns:**

Not applicable.

**Strength And Weaknesses:**

**Strong points.**

The proposed learning problem is interesting. The proposed method is backed by strong theoretical analyses and empirical results. The paper is well-written and easy to understand.

**Weak points.**

The concern I have is that whether the proposed learning problem is a valid weakly supervised learning paradigm. Confidence labels (posterior class probabilities $P(Y=1|X)$) certainly contains more information than the usual labels. On the other hand, unlabeled data pairs and differences somewhat constitute a weaker learning problem. Thus, to me, it is not entirely clear whether learning with unlabeled data pairs and confidence differences is a weaker or stronger learning problem.

Also, in the experiments, Pcomp was compared against. However, as mentioned in Section 2.3, in Pcomp classification, the learner only knows unlabeled pairs, and within each pair $x,x'$, which one is more likely to have positive label than the other (whether $P(y'=1|x') > P(y=1|x)$). Clearly, ConfDiff classification has access to more information. Thus, it is expected that ConfDiff can outperform Pcomp. Nevertheless, it is okay to use Pcomp as a baseline.

**Summary Of The Paper:**

This work proposed a novel learning problem for binary classifications, namely Confidence-Difference (ConfDiff) classification, in which the learner is only given unlabeled data pairs $x,x'$ equipped with confidence difference specifying the difference in the probabilities of being positive $P(y'=1|x') - P(y=1|x)$. The authors proposed to solve the problem via ERM by constructing an unbiased risk estimator. Estimation error bound was derived, and the robustness of the risk estimator was analyzed. Experiments demonstrated effectiveness of the proposed method.

**Summary Of The Review:**

I vote for a weak accept as I think the paper overall meets the standard, and its contributions outweigh its shortcomings.

---

> ### Author Response · Authors · 2022-11-18
> **Response to Reviewer dSMd**
>
> Thank you very much for your valuable comments. We are encouraged that you agree with the advantages and contributions of our paper. Below are the responses to your comments/questions.
>
> ***
>
> **Q1: Whether ConfDiff is a valid weakly supervised learning paradigm.**
>
> **A1:** Thank you very much for your insightful question. We agree with you that accurate class-posterior probabilities may contain more information. ConfDiff classification is a weakly supervised learning problem in real-world applications. First, it is impossible for the labelers to give the ground-truth confidence values for training examples in real-world applications. In the main body of this paper, we analyzed ConfDiff classification with accurate confidence values for generalized analysis. However, We can only collect inaccurate confidence values in real-world applications. We investigate the impacts of noisy confidence difference from both theoretical and empirical perspectives in this paper. Theoretically, we prove that the risk estimator is still consistent when the sum of absolute errors of confidence values has a sublinear growth rate. Empirically, we find the performance stable when the confidence values are very noisy, even with many of their signs flipped. Second, the conceptual baseline in this work is Pconf classification, where pointwise confidence is given for each positive example. We considered the supervision information much weaker than the Pconf classification, given unlabeled data and pairwise confidence difference. Based on these two reasons, ConfDiff classification is a weakly supervised learning problem in real-world applications.
>
> ***
>
> **Q2: It is expected that ConfDiff can outperform Pcomp because ConfDiff has more information.**
>
> **A2:** Thank you very much for your comments. We agree that ConfDiff methods are given more supervision information than Pcomp methods. However, properly leveraging the supervision information of confidence difference has not been studied before our work. The contribution of our work is that we consider the confidence difference for the first time, which can be further obtained when collecting Pcomp data. We propose an effective method with theoretical guarantees for it.
>
> Furthermore, from Subsection 4.3 and Appendix H, we can observe that given only a small fraction (e.g., 10%) of training data, ConfDiff methods can outperform Pcomp methods equipped with the strong consistency regularization technique by a large margin. For example, on Fashion ($\pi_{+}=0.2$), ConfDiff-ABS achieved 0.953 with 10% of the training data, while Pcomp-Teacher achieved 0.858 with 100% of the training data. On Pendigits ($\pi_{+}=0.5$), ConfDiff-ABS achieved 0.949 with 10% of training data, while Pcomp-Teacher achieved 0.878 with 100% of training data. These results elucidate that utilizing confidence difference by our approach may be much more beneficial than increasing training data, which can validate the superiority of our proposed method against Pcomp methods.
>
> Besides, we also add the supervised learning method as a baseline for comparisons. We have access to the ground-truth labels of training data for the supervised learning method, which can be regarded as the oracle method. Due to page limits, we only present the results of Pcomp-Teacher, ConfDiff-ABS, and the supervised learning method on the eight data sets with $\pi_{+}=0.2$ for illustration. We can get similar results with $\pi_{+}=0.5$ and $\pi_{+}=0.8$.
>
> |Method|MNIST|Kuzushiji|Fashion|CIFAR-10|
> |---|---|---|---|---|
> |Pcomp-Teacher|0.965±0.010|0.871±0.046|0.853±0.017|0.836±0.019|
> |ConfDiff-ABS|0.975±0.003|0.898±0.003|0.965±0.002|0.862±0.015|
> |Supervised|0.990±0.000|0.939±0.001|0.979±0.001|0.894±0.003|
>
> |Method|Optdigits|USPS|Pendigits|Letter|
> |---|---|---|---|---|
> |Pcomp-Teacher|0.901±0.023|0.894±0.023|0.928±0.019|0.883±0.006|
> |ConfDiff-ABS|0.963±0.009|0.960±0.005|0.988±0.002|0.942±0.007|
> |Supervised|0.990±0.002|0.984±0.002|0.997±0.001|0.978±0.003|
>
> Based on the results above, we can see that the performance of ConfDiff-ABS is superior against Pcomp-Teacher with the strong consistency regularization technique and comparable with the supervised learning methods in some cases. Therefore, the effectiveness of utilizing the supervision information of confidence difference is validated.

---

### Official Review · Reviewer_sacA · 2022-10-25

**Confidence:** 3
**Clarity, Quality, Novelty And Reproducibility:** Overall the paper is clear and well-w…
**Correctness:** 4
**Technical Novelty And Significance:** 3
**Empirical Novelty And Significance:** 3
**Recommendation:** 6

**Strength And Weaknesses:**

+: the paper is clearly written, and I think the proofs are correct (I am less familiar with generalisation bounds, so I cannot give strong guarantees for Theorem 3 and the next ones, but the reasoning seems ok to me).

+: As far as I know, the proposed framework is original

-: it is not clear at all to me, despite the two high-level examples given in the introduction, that providing accurate probability differences for pairs of data is much easier/simpler than providing single probability estimates. While I would agree that providing qualitative comparison (Pcomp framework) is easier, it is not so clear for probability differences. And the argument put forward later on that different experts may disagree on probabilistic estimates is equally true for probabilistic differences. Do we have actual evidence (and not illustrative example, that I have a hard time really figuring out) that this is indeed the case in some applications? Maybe the authors could elaborate a bit more on their examples to provide a fuller story, where the estimates are explicitly mentioned?

-: while Pcomp and Pconf can be adapted easily to multi-class setting, it seems intuitively harder to adapt the current proposal to more than two classes, both from a theoretical perspective but also from a practical perspective (it seems very demanding and hard to elicit probability differences over all pair of classes). Could some comments be provided with respect to that?

-: I do not understand why the framework is not compared, in Table 1 (or in appendices tables) to both Pconf and the framework where data are fully observed (rather than using various Pcomp variants).

-(minor): I wonder why in the test non-binary data sets were turned into binary data sets. UCI and other repositories contain plenty of native binary data sets, and it is unclear why those particular data sets were chosen, and why the separation was chosen the way it was?

**Summary Of The Paper:**

The paper proposes to learn from observing probability differences between pairs of labels, in a binary classification setting. The premise is that obtaining such information is sometimes easier than obtaining single probability estimate, and richer than simple qualitative comparisons.

The paper demonstrates that such information is sufficient to converge towards the optimal risk, deals with the case of corrupted estimates, and also propose a correction to avoid the loss becoming negative through a ReLu correction.

Some experiments are done, in particular to compare with a qualitative feedback.

**Summary Of The Review:**

The paper proposes a new learning framework considering as supervision differences of probabilities. While the theoretical analysis is interesting, it is unclear how practical this framework is, nor how realistic is the assumption that we can have good estimates of probability differences. It is also unclear how the framework could extend beyond binary classification.

---

> ### Author Response · Authors · 2022-11-18
> **Response to Reviewer sacA (PART 1/2)**
>
> First, we sincerely thank you for your time and efforts in reviewing our submission. Next, we would like to respond to major concerns raised in the reviewing comments.
>
> ***
>
> **Q1: Whether providing accurate confidence difference is easier and simpler and whether there is some actual evidence.**
>
> **A1:** Thank you very much for your question. First, the simplicity of the ConfDiff classification lies in that we only need a confidence value indicating the difference in the probability of being positive for a pair of examples. Second, it is impossible to provide the exact confidence values in real-world scenarios, whether for ConfDiff or Pconf classification. We can only obtain inaccurate confidence values in reality. Fortunately, theoretical and empirical results show that our method can perform well without accurate confidence values. Therefore, the supervision required here is less than the Pconf classification.
>
> We take the short-video recommender system as an application example to demonstrate some potential advantages of confidence difference. In this task, the combination of a user and a video can be regarded as an instance, and the liking/disliking can be considered as the positive/negative class. Here, the level of liking can be reflected by the watching ratio (watch_duration/video_duration). However, pointwise confidence is noisy and biased since different people with different characteristics have different watching habits. For example, some people tend to watch videos for a long duration, while some people tend to watch videos for a short duration. Therefore, considering them all together in the training set may have negative effects. In such cases, confidence difference may be a more objective statistic by describing the relative difference in the interests in different videos for a given user.
>
> We tested different methods on a real-world short-video recommendation data set [1]. Please see more experimental details in A1 in response to Reviewer bMCm.
> ***
> **Q2: Extension to the multi-class setting.**
>
> **A2:** Thank you very much for your excellent suggestion. We can extend ConfDiff classification by adopting the widely used one-versus-all strategy for multi-class classification [2]. Suppose we have $q$ classes for a specific class. We may collect some unlabeled data pairs with confidence difference specifying the difference in the probabilities of this class label being the ground-truth label for these two examples. In this way, we can train a binary classifier for each class. Finally, the predicted label for a testing example can be determined by comparing the outputs of the $q$ binary classifiers. Therefore, extending our work to the multi-class setting is very promising.
>
> ***
>
> **Q3: Comparison of both Pconf and supervised learning methods.**
>
> **A3:** Thank you very much for your helpful suggestions. We agree with you that it will be more convincing by comparing with Pconf and supervised learning methods. We have access to the ground-truth labels of training examples for the supervised learning method. Notably, Pconf only uses positive data. Therefore, to make use of both positive and negative data for a fair comparison, we used the risk estimator in Eq. (4) of [3]. We denote it as Pconf+.
> Here are the experimental results on the four benchmark data sets:
>
> |Class Prior|Method|MNIST|Kuzushiji|Fashion|CIFAR-10|
> |---|---|---|---|---|---|
> |$\pi_{+}=0.2$|Supervised|0.990±0.000|0.939±0.001|0.979±0.001|0.894±0.003|
> |$\pi_{+}=0.2$|Pconf+|0.989±0.001|0.938±0.003|0.979±0.002|0.891±0.007|
> |$\pi_{+}=0.5$|Supervised|0.986±0.000|0.929±0.002|0.976±0.001|0.871±0.003|
> |$\pi_{+}=0.5$|Pconf+|0.985±0.001|0.927±0.001|0.978±0.001|0.877±0.002|
> |$\pi_{+}=0.8$|Supervised|0.991±0.001|0.942±0.003|0.979±0.000|0.897±0.002|
> |$\pi_{+}=0.8$|Pconf+|0.990±0.001|0.945±0.002|0.980±0.001|0.905±0.004|
>
> Here are the experimental results on the four UCI data sets:
>
> |Class Prior|Method|Optdigits|USPS|Pendigits|Letter|
> |---|---|---|---|---|---|
> |$\pi_{+}=0.2$|Supervised|0.990±0.002|0.984±0.002|0.997±0.001|0.978±0.003|
> |$\pi_{+}=0.2$|Pconf+|0.990±0.005|0.983±0.005|0.998±0.001|0.972±0.007|
> |$\pi_{+}=0.5$|Supervised|0.988±0.003|0.980±0.003|0.997±0.001|0.975±0.001|
> |$\pi_{+}=0.5$|Pconf+|0.989±0.003|0.981±0.004|0.997±0.001|0.968±0.004|
> |$\pi_{+}=0.8$|Supervised|0.987±0.003|0.983±0.002|0.997±0.001|0.976±0.004|
> |$\pi_{+}=0.8$|Pconf+|0.988±0.003|0.984±0.004|0.997±0.002|0.968±0.008|
>
> Based on Table 1, Table 2, and the results above, we can see that the performance of ConfDiff-ABS is competitive against the supervised learning method in many cases, which validates the effectiveness of the proposed approach.

---

> ### Author Response · Authors · 2022-11-18
> **Response to Reviewer sacA (PART 2/2)**
>
> **Q4: The choice and partition of data sets.**
>
> **A4:** Thank you very much for your question. We choose the data sets and the corresponding partitions by following [4] completely. Because the compared methods, i.e., the Pcomp methods, perform well on these data sets with the corresponding partitions, we adopt them in this paper. We can observe similar experimental results on other binary-class UCI data sets.
>
> ***
>
> Reference:
>
> [1] KuaiRec: A Fully-observed dataset and insights for evaluating recommender systems, in CIKM 2022.
>
> [2] Statistical analysis of some multi-category large margin classification methods, in JMLR 2004.
>
> [3] Binary classification from positive-confidence data, in NeurIPS 2018.
>
> [4] Pointwise binary classification with pairwise confidence comparisons, in ICML 2021.

---

### Official Review · Reviewer_bMCm · 2022-11-04

**Confidence:** 3
**Correctness:** 2
**Technical Novelty And Significance:** 2
**Empirical Novelty And Significance:** 2
**Recommendation:** 3

**Clarity, Quality, Novelty And Reproducibility:**

The paper is clearly written. As far as I can tell, the reproducibility is high (although I didn't read all the proofs.)

Regarding the novelty, while I do think the ConfDiff setting is new, I am doubtful if it is a realistic or useful setting in practice, as mentioned above.

**Strength And Weaknesses:**

### Strength

- The paper is technically well formulated. The proposed learning setting was rigorously set up and an unbiased risk estimator is derived for the empirical risk minimization.
- The paper is well written and easy to follow. Even people outside this particular field should be able to grasp the general idea proposed therein.

### Weaknesses

- In my opinion, the biggest weakness of this paper is its setting. Is the setting realistic? The paper doesn't provide sufficient motivation in the introduction; it also lacks realistic experimental setup to support the ConfDiff setting.

**Summary Of The Paper:**

This paper studies a weakly supervised learning setting, in which one has limited access to the *confidence labels* of the training examples. Previous work in this line includes the setup with pointwise confidence scores (Pconf), and the setup where pairwise comparisons of the confidence scores are available (Pcomp). This work presumes more fine-grained information than the latter: Unlabeled data pairs with confidence difference (ConfDiff). This learning setup is then formulated as empirical risk minimization and a corresponding unbiased risk estimator is constructed, together with an estimation error bound.



**Summary Of The Review:**

As mentioned earlier, my key issue of this paper is that I don't think the ConfDiff setting is realistic.

In terms of the level of information required in training, ConfDiff sits in between Pconf and Pcomp, in theory. In practice, however, I don't see a situation where one obtains the exact difference between two confidence scores, **without** first estimating the pointwise confidence scores. Even the experiments in the paper have to first do the point estimation.

It was mentioned in the paper that *the confidence difference is given by annotators in real-world applications*. Has such annotation procedure ever actually applied in the real world? We have to realize that it is extremely hard for a human annotator to give exact confidence difference between two examples. I would say the annotation settings of Pconf and Pcomp are, comparably speaking, more realistic. In the case of former, each annotation provides more information. (Also I'd say the annotator might have to do pointwise estimates first before giving exact confidence differences). In the case of latter, the annotation is much simpler for the annotator as it is only a qualitative paired comparison.

In any case, I think the paper should motivate the proposed setting better, ideally with some real-world applications.

---

> ### Author Response · Authors · 2022-11-18
> **Response to Reviewer bMCm**
>
> First of all, we wish to sincerely thank you for your time and efforts in reviewing our submitted paper. Next, we would like to respond to major concerns raised in the reviewing comments.
>
> ***
>
> **Q1: The real-world application of the ConfDiff setting.**
>
> **A1:** Thank you very much for your question. We strongly agree that having a real-world application will strengthen the paper greatly. For the setting, we agree with you that we can estimate the pointwise confidence apart from the confidence difference in many cases. However, many cases still exist when we can't obtain accurate pointwise confidence, such as cases related to privacy or security issues. In these cases,
> we may ask labelers to estimate the difference in the probabilities of being positive. It is worth noting that it is impossible to provide the exact confidence values in real-world applications. Both theoretical and empirical results show that our method can perform well without accurate confidence values. Therefore, this setting may have promising and practical applications in the future. We will consider finding a real-world data set with confidence difference instead of pointwise confidence in our future work.
>
> Besides, when pointwise confidence is available, there are many cases in real-world applications where the confidence is noisy and biased, which will degenerate the performance. Sometimes, considering the confidence difference may be an effective alternative. Here, we found promising real-world applications in recommender systems. Take the short-video recommender system as an instance. In this task, the combination of a user and a video can be regarded as an instance, and the liking/disliking can be considered as the positive/negative class. In these applications, the explicit liking signals may not be accessible, and we can only make use of implicit feedback to "guess" the users' interests. Here, the level of liking can be reflected by the watching ratio (watch_duration/video_duration), which is collected in many related data sets. However, pointwise confidence is noisy and biased since different people with different characteristics have different watching habits. For example, some people tend to watch videos for a long duration, while some people tend to watch videos for a short duration. Therefore, considering them all together in the training set may have negative effects. Notably, these factors can hardly be considered for the model since we may only have access to the interaction history between users and items. In such cases, confidence difference may be a more objective statistic by describing the relative difference in the interests in different videos for a given user.
>
> We tested different methods on a real-world short-video recommendation data set [1]. Due to the time limit, we sampled 1000 high-frequency users and 500 items. The pointwise confidence was generated by dividing the watching ratio by two since most watching ratios lay from zero to two. We adopted the NCF model [2] as our backbone. The compared methods include 1) Supervised: We sampled the label from a Bernoulli distribution parameterized by the pointwise confidence for each example and regarded it as the ground-truth label. Then we trained the model with the cross-entropy loss; 2) Pcomp-Teacher: The most competitive Pcomp method in the paper; 3) Pconf+: A method minimizing the risk estimator in Eq. (4) of [3], using the pointwise confidence of unlabeled data; 4) ConfDiff-ABS: The most competitive ConfDiff method in the paper. We used the default hyperparameters and experimental protocols in [2] for all the methods and reported the hit ratio (HR) and normalized discounted cumulative gain (NDCG) results.
>
> |Method|HR|NDCG|
> |---|---|---|
> |Supervised|0.224|0.104|
> Pcomp-Teacher|0.293|0.120|
> Pconf+|0.430|**0.225**|
> ConfDiff-ABS|**0.466**|**0.225**|
>
> Based on the results, we can observe that ConfDiff-ABS performs better than Pconf+ in terms of HR and performs comparably against Pconf+ in terms of NDCG. Therefore, the effectiveness of our methods in the noisy confidence setting is validated. We plan to report the experimental results with larger data sets and more tasks (e.g., news recommendations) in the next version of our paper.
>
> ***
> Reference:
>
> [1] KuaiRec: A fully-observed dataset and insights for evaluating recommender systems, in CIKM 2022.
>
> [2] Neural collaborative filtering, in WWW 2017.
>
> [3] Binary classification from positive-confidence data, in NeurIPS 2018.

---

### Author Response · Authors · 2022-11-28
**General Response**

First, we sincerely thank all the reviewers for their time and efforts in reviewing this submission. We are encouraged that they think the investigated problem is novel (Reviewer sacA and Reviewer dSMd), and the theoretical analysis is interesting and solid (Reviewer sacA and Reviewer dSMd). Besides, we are glad that all the reviewers think the manuscript is well-written and easy to follow.

We respond to each reviewer in detail. We will revise the manuscript according to the reviewers' suggestions in the next version, and the main changes will include the following:
- More detailed descriptions of our motivation in the Introduction, including the motivation from real-world applications such as the application for the short-video recommender system task.
- Comprehensive experimental results on real-world applications, such as the complete experimental results on more large-scale data sets of short-video recommender systems.
- Experimental results of the supervised learning method on all the benchmark data sets.
- More discussions of the problem setting, such as the reasons for being a weakly supervised learning problem, the extension to the multi-class setting, and so on.

---

### Decision · Program_Chairs · 2023-01-20

**Decision:**

Reject

**Justification For Why Not Higher Score:**

The authors fail to provide a better motivation for the setup, which henceforth seems a bit artificial.


**Justification For Why Not Lower Score:**

The technical material is correct and the write-up is good.

**Metareview: Summary, Strengths And Weaknesses:**

This paper addresses a learning setting where data come as unlabeled pairs decorated with the difference between their probability for each of them to be positive data, a setup called ConfDiff. The authors provide unbiased estimators of the risk (depending on the loss functions) that are used to derive learning algorithm.

+ paper well written
+ the topic of getting away from the traditional labelled training data is key, as getting labels may be expensive
+ the used technique: recovering an unbiased estimator of the risk and then minimizing it is relevant

- the motivation and applicability of the learning setting are not obvious
- the used technique is a "traditional" technique encountered when dealing with noisy, positively-unlabeled, or confidence-rated labels

The authors haven't discussed "structural" results as those coming from the seminal PAC literature making the connection between learnable classes of concepts like those learnable under classification noise, statistical queries, class-conditional classification noise, constant-partition classification noise... Those are general results making reductions from one "non-usual" setup of learning to the other (see Kearns, 1998, Blum, 2000, Ralaivola et al, 2006).